# Rapid and on-site wireless immunoassay of respiratory virus aerosols via hydrogel-modulated resonators

Xin Li[1,2,6], Rujing Sun[1,3,6], Jingying Pan[1,4], Zhenghan Shi[1], Zijian An[1], Chaobo Dai[1], Jingjiang Lv[1], Guang Liu[1], Hao Liang ®[3], Jun Liu[1,2], Yanli Lu ®[1,5], Fenni Zhang[1] & Qingjun Liu ®[1,2] ✉

Rapid and accurate detection of respiratory virus aerosols is highlighted for virus surveillance and infection control. Here, we report a wireless immunoassay technology for fast (within 10 min), on-site (wireless and battery-free), and sensitive (limit of detection down to fg/L) detection of virus antigens in aerosols. The wireless immunoassay leverages the immuno-responsive hydrogel-modulated radio frequency resonant sensor to capture and amplify the recognition of virus antigen, and flexible readout network to transduce the immuno bindings into electrical signals. The wireless immunoassay achieves simultaneous detection of respiratory viruses such as severe acute respiratory syndrome coronavirus 2, influenza A H1N1 virus, and respiratory syncytial virus for community infection surveillance. Direct detection of unpretreated clinical samples further demonstrates high accuracy for diagnosis of respiratory virus infection. This work provides a sensitive and accurate immunoassay technology for on-site virus detection and disease diagnosis compatible with wearable integration.

Continuing evolution of SARS-CoV-2 (severe acute respiratory syndrome coronavirus 2) together with seasonal outbreak of infectious respiratory viruses such as influenza and RSV (respiratory syncytial virus) have imposed a heavy burden on public healthcare[1–3]. To counter the threaten from evolving respiratory viruses in the post-pandemic era, developing fast and accurate diagnostic tools has been and will always be one of the priorities in the field. Particularly, respiratory virus can be spread by the airborne route in the form of small droplet (<5 μm), namely aerosols, which can remain suspended in environment and contagious for hours[1,3]. For instance, study shows that aerosols originate from human respiration behaviors such as talking, coughing, and sneezing, etc. are the key medium responsible for the wide spread of COVID-19[4]. However, exhaled aerosols contain a complexed biological matrix rich in proteins, nucleic acids, and volatile organic compounds[5,6], combined with greatly varied viral load of individuals, which lead to challenges in accurate on-site detection of virus aerosols.

Nucleic acid testing[7–10] and immunoassay[10–12] offer powerful tools to counter spread of infectious respiratory diseases by providing point-of-care virus surveillance. Recent advances have realized viral RNA or antigen detection in aerosols via CRISPR (clustered regularly interspaced short palindromic repeats) tools[13], transistors[14], and electrochemical devices[15,16], respectively. However, limitations including tedious pretreatment and amplification, lack of validation in clinical samples, complexed sample collection and transfer, or inadequate sensitivity hinder its medical translation. Therefore, what has remained

[1]Biosensor National Special Laboratory, Key Laboratory for Biomedical Engineering of Education Ministry, Department of Biomedical Engineering, Zhejiang University, Hangzhou 310027, China. [2]Taizhou Key Laboratory of Medical Devices and Advanced Materials, Research Institute of Zhejiang University-Taizhou, Taizhou 318000, China. [3]Guangxi Key Laboratory of AIDS Prevention and Treatment, School of Public Health, Biosafety III Laboratory, Life Science Institute, Guangxi Medical University, Nanning 530021 Guangxi, China. [4]School of Medicine, Zhejiang University, Hangzhou 310027, China. [5]Intelligent Perception Research Institute, Zhejiang Lab, Hangzhou 311100, China. [6]These authors contributed equally: Xin Li, Rujing Sun. ✉e-mail: qjliu@zju.edu.cn

unexplored so far is a rapid, accurate, and on-site virus aerosol detection technology with robust readout and clinical validation, and ideally with wireless and battery-free configuration to be compactly integrated in daily wearables.

Recent years have witnessed the rise of hydrogel metamaterials in biomedical sensing applications, owing to their programmable structure and responsiveness to wide range of physical or biochemical stimuli[17,18]. Despite bio-recognition elements such as antigen/antibody, enzymes, nucleic acids have been incorporated into hydrogels for specific bio-responsiveness, colorimetric or mechanical responses (e.g., volume swelling) of conventional hydrogels provide inadequate sensitivity to detect scarce biomolecules[19–21]. To amplify the signals, hydrogel metamaterials can work as building blocks to couple with neighboring optical or electrical transducers[22,23]. Particularly, radio frequency (RF) resonators are featured with delicate inductance/ capacitance structures which can be modulated by mechanical or dielectric variations, thus providing ideal transducers to exploit the responsiveness of hydrogel metamaterials[23–25]. The wireless readout and compatibility with wearable integration also allow for miniaturization and in situ signal recording, which make RF sensors suitable for advanced healthcare monitoring[26–28].

Here, we report a wireless immunoassay technology based on immuno-responsive hydrogel-modulated resonant (named as ImmHR) sensors and RF readout network, demonstrating rapid virus aerosol detection (<10 min) with limit of detection (LoD) down to fg/L or sub-fg/L levels. To realize ultrasensitive and specific antigen detection, antibody modified gold nanoparticles (AuNPs-Ab) are immobilized within the hydrogel via immuno crosslinks between AuNPs-Ab and grafted antigen on polymerized acrylamide (AAm) chains. These reversible immuno crosslinks are locally focused around AuNPs-Ab, which can be broken upon antigen stimuli (virus aerosols), rapidly release the strain energy and trigger dramatic hydrogel swelling. The incorporation of intermediate relay (IR) coils, on the other hand, enables a paralleled RF readout network that facilitate robust and multiplexed detection of respiratory virus in a wearable manner. As a demonstration, the wireless immunoassay technology manifests sensitive and specific detection of SARS-CoV-2, influenza A H1N1, and RSV, for wearable and on-site monitoring of certain virus infection or co-infection of multiple viruses. The wireless immunoassay also shows high accuracy in surveillance of clinical samples, offering a facile, accurate, and versatile platform for biosensing and diagnosis of contagious respiratory diseases.

## Results

### Structure and mechanism of wireless immunoassay technology

The wireless immunoassay technology consists of ImmHR sensors and a RF readout network. To boost the capability and flexibility for detection of various specimen originated from respiratory tract (saliva, aerosols, etc.), the ImmHR sensor couples an immuno-responsive hydrogel and a pair of RF resonators, creating a tunable resonant structure that transduces the stimuli of target antigen into resonant response (Fig. 1a). The immuno-responsive hydrogel is constructed by grafting the vinyl-modified antigen and corresponding antibody conjugated AuNPs (AuNPs-Ab) into polymerized AAm (Supplementary Fig. 1), so that their interactions introduce immuno crosslinks in the network. Notably, the vinyl-antigen demonstrate an order of magnitude higher dissociation equilibrium constant ($K_D$) towards antibody compared with pristine antigen ($4.2 \times 10^{-8}$ versus $2.4 \times 10^{-9}$, detailed in Supplementary Fig. 2a–c). Thus, competitive binding of pristine antigen (virus stimuli) can break these immuno crosslinks and disrupt the preformed hydrogel network, as indicated by the lower storage modulus after immunoassay (Supplementary Fig. 3). Meanwhile, the abundant binding sites on virus particles lead these AuNPs-Ab to bind together and form aggregates (Supplementary Fig. 4a, b and Supplementary Fig. 9c). Due to the LSPR effect, the blue color of hydrogels

gradually fades out (Supplementary Fig. 4c). As the hydrogel serves as the dielectric interlayer of the ImmHR sensor, the swelling of which directly modulates the capacitance of the resonator, and subsequently shifts the resonant frequency (Fig. 1b). The adoption of paired RF resonators is favored for the merits of flexible geometry, tunable resonant frequency, and localized electric field between two resonators that limits the external environment disturbance[23].

Compared with two-dimensional (2D) sensing interfaces, the three-dimensional (3D) hydrogel network provides inherent porous structure and high specific surface area for analyte diffusion and capture[20]. The network of immuno-responsive hydrogel exhibited expansion after 3-h-of antigen stimuli in 1 ng/mL antigen spiked solution (Fig. 1c and Supplementary Fig. 5a, b). Specifically, antigen stimuli induced faster swelling rate and higher swelling ratio compared to that in deionized water, which reached equilibrium after 48 h, respectively (Supplementary Fig. 5c). The faster swelling rate in antigen spiked samples can be attributed to the break of immuno crosslinks that additionally decreased the overall crosslink intensity that facilitated faster water molecules absorption, which was lacking in DI water. When reaching to equilibrium swelling, the swelling ratio in 1 ng/mL antigen was also higher than that in DI water, the results corresponded well with the designed mechanism that upon stimuli from virus antigens, the hydrogel network would be disrupted more significantly and induce higher swelling ratio. Besides, the facile assembly process also enables scalable manufacturing of the ImmHR sensors (Supplementary Fig. 6). The thin and miniaturized ImmHR sensors can be further programmed into different resonant frequencies in forms of reversely aligned split rings and spiral coils (Fig. 1d and Supplementary Fig. 7), since capacitances of these resonators can be tuned via geometry design and interlayer adjustment.

To form the RF readout network for wireless and multiplexed immunoassays, ImmHR sensors, IR coils, and a read coil are assembled and wirelessly coupled (Fig. 1e). As a demonstration, three ImmHR sensors targeted for contagious respiratory virus (SARS-CoV-2, influenza A H1N1, and RSV) and a temperature sensor for monitoring of breath temperature have been integrated, which aims to provide fast virus discrimination and monitoring of potential fever symptom. In this structure, the IR coils play the vital role as paralleled communication paths not only bounded to individual ImmHR sensors but also wirelessly coupled to the read coil (Supplementary Fig. 8a). The flexible IR coils facilitate the propagation of electromagnetic field to individual sensors along its pathlength or through designed inductive terminals[28], enhancing the mechanical robustness of the network and stabilizing the multiplexed RF readout when it undergoes misalignment or deformation. The read coil, on the other hand, is a one-port planar coil connected to the portable vector network analyzer (VNA), which enables electromagnetic field emission and return loss ($S_{11}$) probing. Thus, each ImmHR sensor is modulated by the capacitance change induced by the responsive hydrogel, coupled with matched IR coils and then forms paralleled electromagnetic connection with the read coil (Supplementary Fig. 8b). With separate resonant peaks that corresponds to individual ImmHR sensors, multiplexed readout of ImmHR sensors can be realized by RF spectrum analysis via a single scanning.

### Hydrogel formation and composition optimization

Here we used SARS-CoV-2 nucleocapsid protein (NP) and corresponding antibody as the model antigen and antibody to prepare immuno-responsive hydrogels. The antigen has been chemically modified by N-succinimidylacrylate (NSA) to introduce vinyl groups, which enables copolymerization with AAm monomer to produce polymerized antigen (Fig. 2a). The crosslinking is realized with N, N'-methylenebisacrylamide (MBAA) and AuNPs-Ab, which serve as covalent crosslinker and immuno crosslinker, respectively. The incorporation of AuNPs-Ab into the hydrogel network forms focused

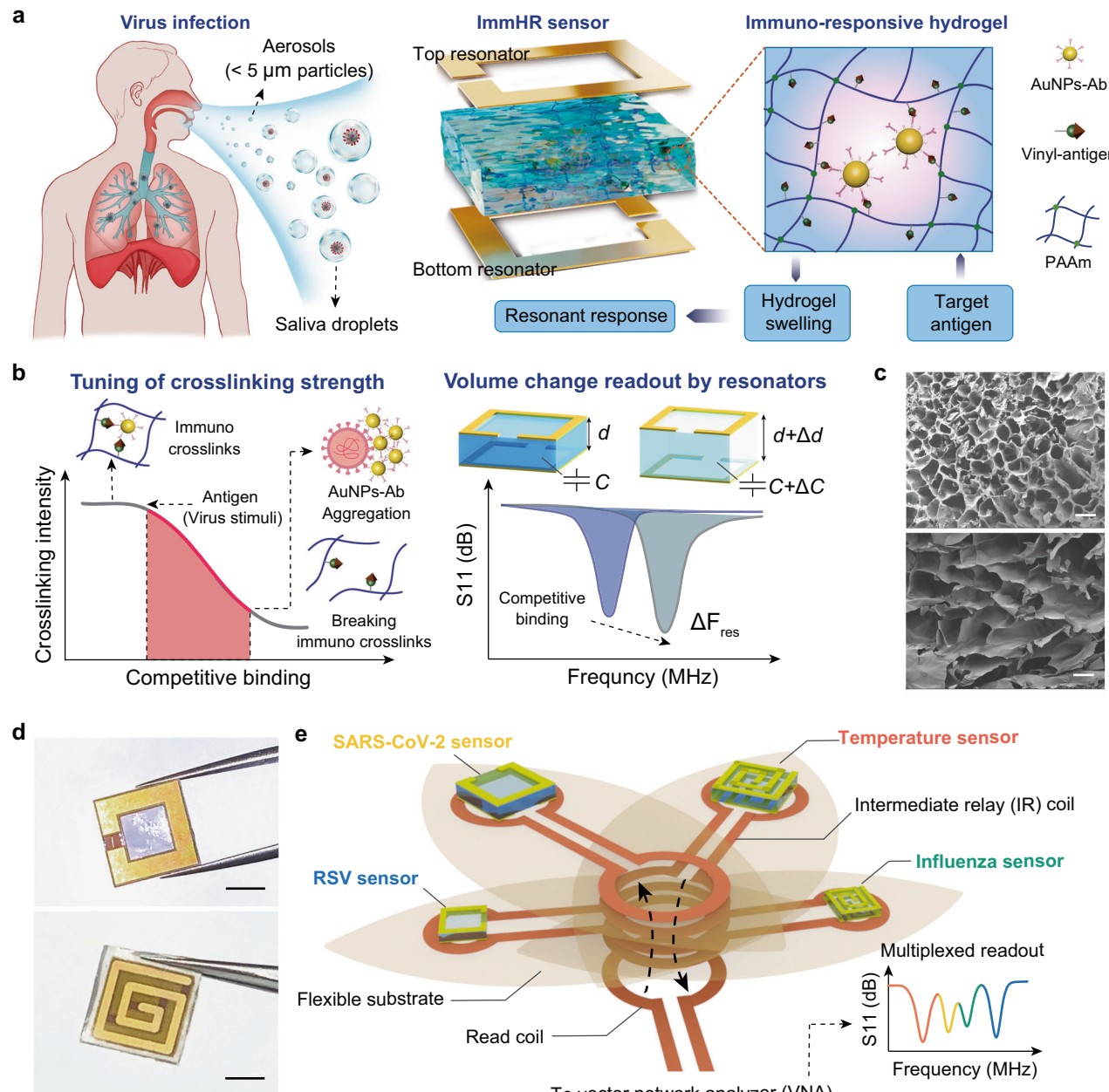

**Fig. 1 | Wireless immunoassays for virus aerosol detection. a** Structure of ImmHR sensor for respiratory virus detection in forms of aerosol (<5 μm particles), and saliva droplets (large particles and liquids). **b** Mechanism of hydrogel immuno response and transduction of electrical signal. **d** thickness of the hydrogel. C, capacitance of the paired RF resonators. $F_{res}$, resonant frequency. **c** Scanning electron microscopy (SEM) images of hydrogels before (top) and after immersing in 1 ng/mL spiked solution for 3 h. Scale bar: 100 μm. The experiments are repeated three times with similar results. **d** Photos of assembled ImmHR sensors in forms of split rings (top) and spiral coils (bottom). Scale bar: 5 mm. **e** Multiplexed readout of ImmHR and temperature sensors for respiratory virus discrimination and infection monitoring.

immuno crosslinks around AuNPs, conferring antigen specificity to capture and concentrate virus antigens for amplified response. Further competitive immuno binding between the virus antigen and AuNPs-Ab triggers intense hydrogel swelling owing to the breaking of immuno crosslinks. The multiple protein-protein and protein-AuNPs interactions such as hydrogen bond, electrostatic interaction, and gold-sulfur interaction also contribute to the assembly of AuNPs-Ab when they dissociate from the polymer chains and form aggregates. Notably, the formation of immuno responsive hydrogel needs a precise two-step control: the first step is a quick copolymerization of AAm and vinyl-antigen with APS and TEMED within 60 s, while the second step is crosslinking of the polymerized antigens with MBAA and AuNPs-Ab within 5 min to realize solidified hydrogels. Notably, inadequate

copolymerization would fail to immobilize AuNPs-Ab in the following crosslinking (it produces AuNPs-Ab-antigen precipitate before the hydrogel formation), while excessive copolymerization would lead to direct formation of hydrogels free of immuno crosslinks.

Due to the steric hindrance of AuNPs-Ab, porous structure at nanoscale has been identified in the immuno-responsive hydrogel (Fig. 2b), in contrast to the smooth morphology of pure poly-acrylamide (PAAm) hydrogel (Supplementary Fig. 9a). These nano-pores are polydisperse with average size of ~72 nm, which swelled to ~117 nm or collapsed after competitive antigen binding (Fig. 2b and Supplementary Fig. 9b). Aggregated AuNPs can be identified at the bottom of hydrogel via element mapping (Supplementary Fig. 9c). This transition of AuNPs dispersity also adds colorimetric response to

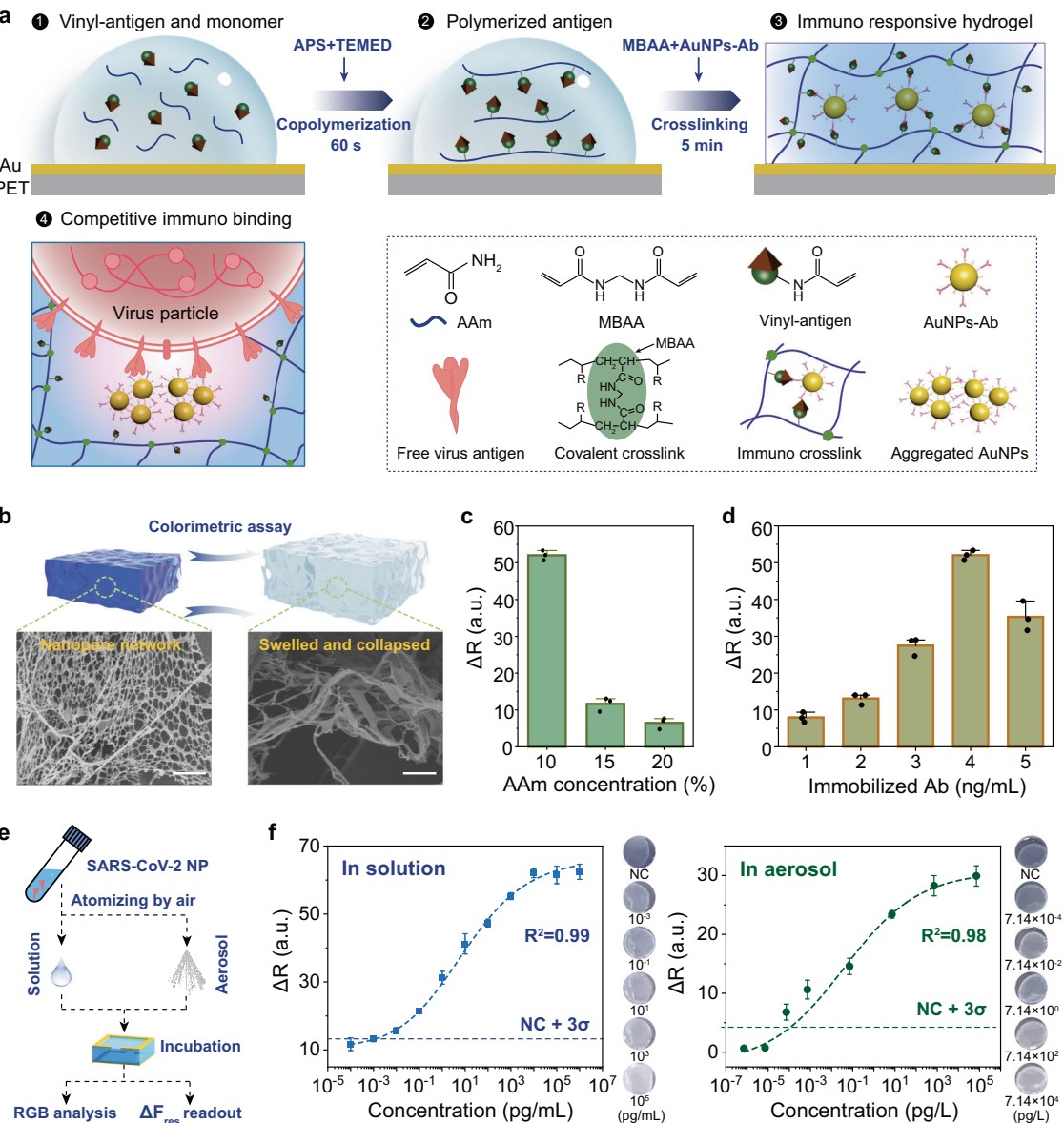

**Fig. 2 | Formation of immuno-responsive hydrogels and optimized immuno response. a** Schematic illustration of immuno-responsive hydrogel preparation. **b** Illustration of colorimetric assay and SEM images of nanopores in hydrogels before/after 1 ng/mL antigen stimuli. Scale bar: 500 nm. The experiments are repeated three times with similar results. **c**, **d** Tuning of AAm (**c**) and immobilized Ab (**d**) concentrations to optimize hydrogel's immuno response, which is evaluated by colorimetric assay using the change of red parameter (ΔR). Error bars are displayed as mean ± σ ($n = 3$ independent experiments), tested in 1 ng/mL antigen.

**e** Illustration of the colorimetric and wireless immunoassay in antigen spiked solution/aerosols. **f** Fitting curves of SARS-CoV-2 NP colorimetric assay in spiked solution (1 × PBS, pH = 7.4) and aerosols (generated from spiked solution, details in Methods) after 90 min incubation, respectively. Right panel shows the color of hydrogels incubated in varied concentration of SARS-CoV-2 NP. The dash lines mark the LoD, determined as the mean response plus 3 σ of NC (negative control). Signals above the LoD are considered distinguishable with >99% confidence. Error bars are displayed as mean ± σ ($n = 3$ independent experiments).

the hydrogel sensor, due to its modulation of localized surface plasmon resonance (LSPR) effect[29,30]. Thus, a facile colorimetric assay can be used to assess hydrogel's immuno response and optimize the composition.

Given that the hydrogel's swelling stems from tuning of crosslink density, both covalent and immuno crosslinking strength are critical, which are determined by AAm and AuNPs-Ab concentration, respectively. The as-prepared hydrogels demonstrate blue color, which gradually fades out due to AuNPs-Ab aggregation in the presence of antigen or virus stimuli. Thus, the colorimetric assay has been used to evaluate the responsiveness of the hydrogels. Images of hydrogels before/after reaction were obtained by a camera. Although the red, green, and blue parameters of the Image J processed hydrogel images

showed similar trend after reaction, the change of red parameter (ΔR) presents a lower standard deviation (σ) and thus been selected for further study (Supplementary Fig. 10). As AAm concentration increases, the ΔR dramatically decreased from 52.0−6.4 in the presence of 1 ng/mL antigen (Fig. 2c). Difference of ΔR can be attributed to the more intense covalent crosslinking that produces stiffer hydrogels, which limits molecule diffusion and hydrogel swelling[31,32]. However, AAm concentration below 10% mass ratio makes it challenging to reach gelation, thus proper covalent crosslinks are necessary to maintain the hydrogel network. Immobilized AuNPs-Ab (evaluated in the concentration of Ab), on the other hand, demonstrates peak ΔR response at 4 ng/mL in hydrogel, indicating excessive immuno crosslinks could have adverse impact on sensitive detection of virus antigen (Fig. 2d).

To explore hydrogel's colorimetric assay of virus antigen, SARS-CoV-2 NP has been spiked in solution and aerosols, respectively (Fig. 2e). Although it took hours to reach equilibrium response in SARS-CoV-2 NP spiked solution, distinct ΔR signal can be measured within 5-10 min (Supplementary Fig. 11). After incubation, the hydrogels manifested obvious color gradient under wide range of antigen stimulation from $10^{-3}$–$10^5$ pg/mL (Fig. 2f), and good linearity can be acquired from $10^{-1}$–$10^4$ pg/mL (Supplementary Fig. 12). Besides, distinguishable ΔR signal can be evoked by liquid samples as low as 10 µL, which facilitates detection of saliva or sneezed droplet from respiratory tract (Supplementary Fig. 13).

Meanwhile, a medical-grade nebulizer was used to produce <5 µm aerosol particles (Supplementary Fig. 14), similar to those occur naturally from human respiration[3]. The concentration of generated aerosol has a conversion factor of 0.0714 from pg/mL in solution to pg/L in aerosol (details in Generation of antigen spiked aerosols, Methods). Compared with solution samples, antigen aerosols induce lower ΔR signal with linear response between $10^{-5}$ – $10^3$ pg/L (Supplementary Fig. 12), yet the non-specific swelling and σ in negative control (NC) have been reduced. To exclude the influence of non-specific swelling due to water absorbance, the LoDs are determined as mean response plus 3 σ of NC. Thus, the LoDs of SARS-CoV-2 NP calculated from the fitting curves are 1.2 fg/mL in solution and 0.18 fg/L in aerosol, respectively.

## Wireless aerosol detection by ImmHR sensors

Resonators such as spiral coils and split rings have been shown to be sensitive transducers to couple physical or chemical responsive materials into wireless detection technology[24,26,33]. However, direct coupling between an resonant sensor and a read coil witnesses $F_{res}$ shift upon relative displacement (Supplementary Fig. 15).The resonant frequency shift is mainly due to the variation of parasitic capacitance during the displacement of two closely approached circuit component[33,34]. The parasitic capacitance is inevitable when designing biosensor based on wireless coupling effect. To handle this misalignment disturbance, ImmHR sensors are fixed on flexible IR coils that magnetically couple to the read coil (Supplementary Fig. 16a–c), which helps stabilize the spectral readout during misalignment between the readout coil and the network[28]. The adoption of IR coils is simple to maintain due to the flexible nature of the IR. Such flexible, grounded magnetic coupling has been utilized to suppress the parasitic capacitance previously[26,27]. Due to the use of IR coils, the impact of misalignment effect has been eliminated, resulting in a stable $F_{res}$ readout regardless of the amplitude of return loss (Fig. 3a).

One possible solution to further improve the signal readout is to enhance the emission field with larger read coils and higher power. Note that the read coil we used is a homemade planar PCB coil, which can be optimized by fabricating with copper spiral wires for higher inductance. The stronger field emission enables better energy harvesting by the sensor network, thus wireless readout can be realized at longer working distances, as demonstrated recently[35]. Influence or sensor displacement, however, can be further suppressed by better grounding design, for instance, addition of another layer of aluminum foil on the back side of the IR coils[27]. In addition, the readout network remains stable under deformation such as bending or twisting (Fig. 3b). The $F_{res}$ of sensors on surfaces with bending radii analog to wearable devices (for instance, a curved face mask) shows little variation, further demonstrating the robustness of the RF readout network (Supplementary Fig. 17).

The modulation effect of hydrogel thickness on $F_{res}$ shift has been validated, as ImmHR sensors with thinner hydrogel films (~500 µm) exhibits lower resonant frequency (268.2 MHz) (Fig. 3c). Although the resonant frequency of as-prepared ImmHR sensors follows the Gaussian distribution due to manufacturing variation, the reproducibility of resonant response can be improved by normalization, defined as $\Delta F_{res}/F_{res}$ (Supplementary Fig. 18). On this basis, thinner hydrogels yielded higher resonant response in the presence of antigen stimuli (Supplementary Fig. 19). However, thinner hydrogels are more susceptible to manufacturing fluctuations, possibly due to the easier dehydration effect during the process, undermining their sensing performances. Thus, ImmHR sensors with 700 µm thick hydrogel are determined for reliable immunoassays in following study, stored in moisture environment (98% RH in refrigerator) to avoid drying.

The resonant responses of ImmHR sensors with PAAm hydrogel, antibody crosslinked hydrogel, and AuNPs-Ab crosslinked hydrogel have been investigated, respectively (Fig. 3d). Due to the lack of bio-recognition elements, the responses of PAAm sensors are independent of the antigen concentration. Antibody crosslinked hydrogels, although endowed antigen responsiveness to the sensors, demonstrated minor resonant response to antigen stimuli (5.3% in 1 ng/mL versus 3.5% in negative control). By contrast, ImmHR sensors with AuNPs-Ab as immuno crosslinker showed 12.9% resonant response to 1 ng/mL antigen. As revealed by rheological tests, the storage modulus (G') of AuNPs-Ab crosslinked hydrogel is much lower than that of antibody crosslinked ones (~80 Pa compared with ~300 Pa, Supplementary Fig. 20a, b), which indicates a more elastic network that facilitates hydrogel swelling and resonant signal transduction.

To assess the ImmHR sensors as versatile platform for virus aerosol detection, SARS-CoV-2 NP, influenza A H1N1 hemagglutinin (HA), as well as respiratory syncytial virus (RSV) fusion protein (FP) have been spiked in phosphate buffered saline (1 × PBS, pH = 7.4) and atomized by nebulizer to generate aerosols with gradient antigen concentration. The resonant signal can be divided into two parts: from non-specific swelling (inherent swelling in the absence of pristine antigen) and from specific swelling caused by competitive binding. To rule out the influence of non-specific swelling, negative control groups (aerosols generated from 1 x PBS with no antigen, termed as NC) has been added to all the experiments as comparison. Considering the existence of the non-specific swelling, the sensor response in NC is regarded as the baseline. The threshold of minimum detectable response is set to be mean response in NC plus three folds of its deviation, as indicated by the dash lines in Fig. 3e. Signals above this threshold are considered as antigen positive, and the limit of detections were calculated from the calibration plots. Resonant responses at 10 min after aerosol exposure are fitted, which indicates LoD of 19.7 fg/L, 0.11 fg/L, and 3.2 fg/L for SARS-CoV-2 NP, H1N1 HA, and RSV FP detection, respectively (Fig. 3e, and unnormalized response curves in Supplementary Fig. 21). The superior sensitivity and low LoD of ImmHR sensors can be further enhanced by prolonging incubation time to 30 min, which demonstrate LoDs of 0.52 fg/L and 0.031 fg/L for SARS-CoV-2 NP and H1N1 HA, respectively (Supplementary Fig. 22). Despite the ImmHR sensors need longer time to reach higher equilibrium response, a detection time within 10 min is able to produce reliable discrimination of virus antigen down to fg/L levels (Fig. 3f). Even more rapid discrimination between spiked aerosols and negative control can be achieved as fast as within 5 min, despite the resonant response is less significant (Supplementary Fig. 23). However, it indicates the high efficiency of this method in virus aerosol detection.

Accompanied with three ImmHR sensors, a temperature responsive resonant sensor has also been exploited to detect the temperature of exhaled breath as an indicator of fever symptom results from respiratory infection. Here, PEG-1500 has been used as temperature sensitive interlayer encapsulated by Ecoflex, which defines the interlayer thickness and prevents liquid leaking[28]. The temperature sensor demonstrates a linear resonant response between 25-45 °C ($R^2$ = 0.99), which covers the variation caused by human breath (Fig. 3g). Besides, the rapid temperature response also enables real-time and reversible detection of breath temperature, making it a feasible tool for wireless monitoring of potential fever (Fig. 3h).

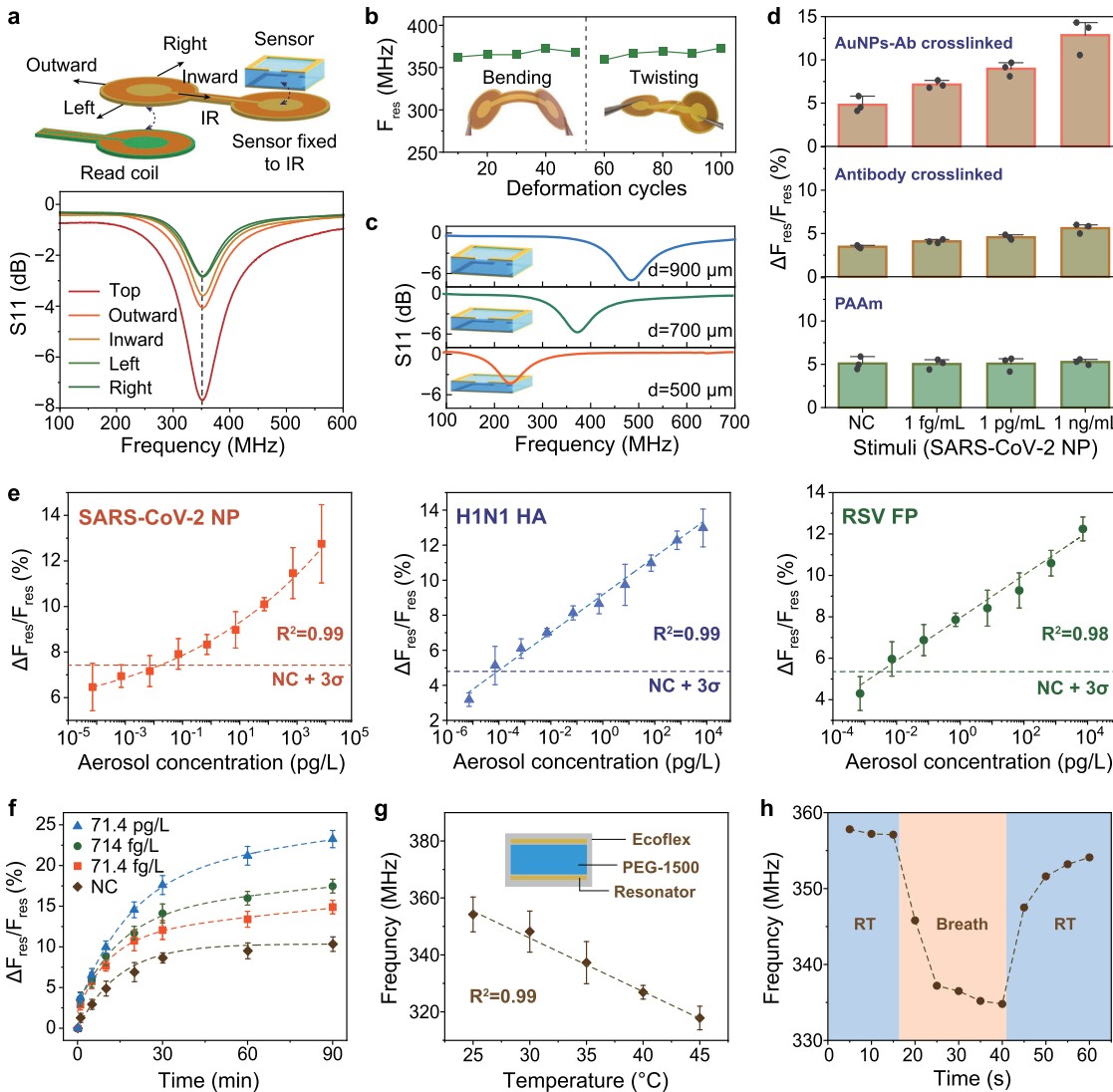

**Fig. 3 | Aerosol detection based on ImmHR sensors. a** Misalignment tests between IR and read coil, including left, right, inward, outward displacement of 2 mm, respectively. **b** Influence of bending and twisting of IR on $F_{res}$ readout. **c** Modulation of hydrogel thickness on $F_{res}$ shift. **d** Resonant response of ImmHR sensors with PAAm hydrogel, antibody crosslinked hydrogel, and AuNPs·Ab crosslinked hydrogels. Error bars are displayed as mean ± σ ($n = 3$ independent experiments). **e** Calibration plot of resonant response versus SARS-CoV-2 NP, H1N1 HA, and RSV FP concentration in spiked aerosol, respectively. The results were acquired at 10 min with 1 × PBS as aerosol matrix. Error bars are displayed as mean ± σ ($n = 3$ independent experiments). **f** Resonant response to SARS-CoV-2 NP spiked aerosols with extended incubation time. Error bars are displayed as mean ± σ ($n = 3$ independent experiments). **g** Calibration plot of the wireless temperature sensor from 25 ~ 45 °C. Inset shows the structure of the temperature sensor. Error bars are displayed as mean ± σ ($n = 3$ independent experiments). **h** Real-time response of the temperature sensor in RT (blue) and breath (yellow). RT, room temperature.

## Wearable integration and multiplexed readout

Wearable breath analysis of volatile organic compounds and aerosols has gained dramatical attention due to the non-invasive sample collection and abundant biomarkers embedded that correlates to diseases or abnormal metabolism[6]. The ImmHR sensor proposed in this work features with combined advantages of rapid detection, superior sensitivity, as well as fully wireless and battery-free configuration. Thus, the miniaturized and lightweight ImmHR sensor array can be fixed on paralleled IR coils and adhered to the surface of a face mask for wearable aerosol detection. Such configuration of wearable sensors that wirelessly coupled to a shared detector (a read coil connected to vector network analyzer) can be broadly deployed for on-site immuno diagnostics, such as at the entry into workplace for community infection surveillance (Fig. 4a and Supplementary Movie 1). The sensor array can be oriented on inside of the face mask for collection and detection of exhaled aerosols. Despite that the ImmHR sensors demonstrate about 200-600 MHz resonant peaks, the integrated face

mask device only works by approaching it (similar to a near field communication (NFC) card) to the reader. The wireless link between the Face Mask Card and the reader establishes within several millimeters (Fig. 4b). For passive devices like the Face Mask Card, the power required to achieve a reliable RF link is only 3-5 mW[35], which makes it rather safe to satisfy the Specific Absorption Rate limit of 1.6 mW/g averaged over one gram of tissue in the head."

As multiple ImmHR sensors are accommodated in the paralleled readout network, the cross-coupling effect among sensors has been studied. By adjusting dielectric thickness and geometry, ImmHR sensors configured at separate $F_{res}$ have been added onto the readout network (Fig. 4c). Despite new new resonant peaks arose in RF spectrum upon sensor addition, the $F_{res}$ of previously loaded ImmHR sensors have barely changed. Meanwhile, the modulation of environment temperature only shifted the response of temperature sensor by −15.4 MHz, while the other three ImmHR sensors remained stable. The neglectable cross-coupling effect can be attributed to the minimal

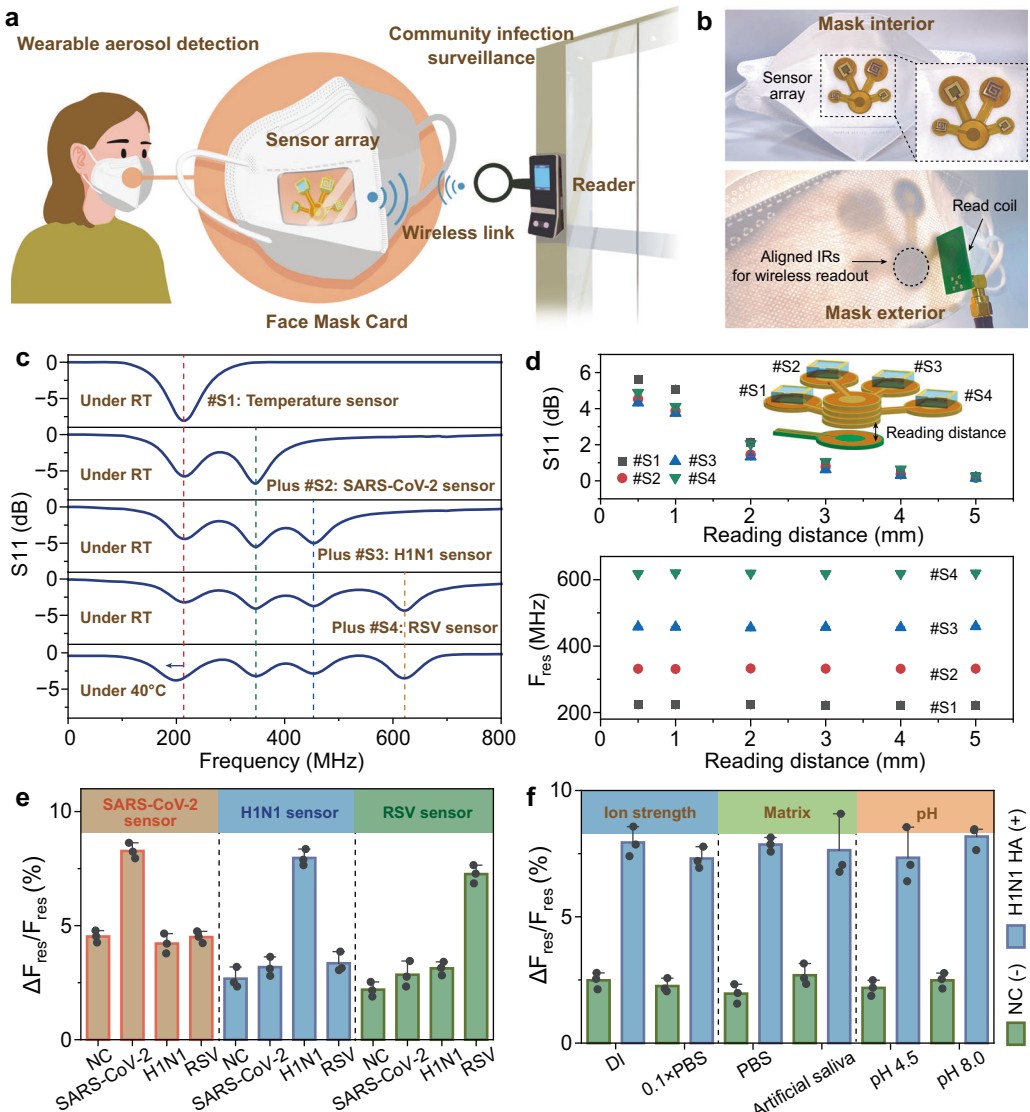

**Fig. 4 | IR-mediated multiplexed sensor readout. a** Illustration of wearable aerosol detection with the ImmHR sensor array for respiratory virus surveillance. The integrated face mask device works similarly to near field communication (NFC) devices, which responds when approaches to the reader device. **b** Photos of the face mask integrated with ImmHR sensors for wireless aerosol detection. **c** RF spectrum measurement upon multiple sensor addition and influence of temperature modulation. **d** Influence of reading distance between IRs and a read coil on multiple sensor readout. **e**, **f** Resonant responses of ImmHR sensors to different respiratory virus antigens (**e**) and biochemical interferences (**f**) with NC as references. Error bars are displayed as mean ± σ (*n* = 3 independent experiments).

mutual inductance among sensors, which are coupled only to the IR coils but not each other[28]. Although prolonged reading distance between IR coils and read coil further decreased the amplitude of $S_{11}$ signals, the $F_{res}$ of all ImmHR sensors remained unaffected (Fig. 4d). However, to ensure a reliable distinction of resonant peaks, the reading distance is confined within 3 mm, which guarantees at least −1 dB amplitude for proper RF spectral measurement. Besides, the integration of the ImmHR sensors and IR coils in a face mask demonstrates similar air permeability to those conventional face masks (Supplementary Fig. 24). Apart from the air permeability, the wireless immunoassay device is thin and lightweight (<1.5 mm thick and 10 g weight), properly adhered and encapsulated on the inner surface of the face mask by medical tapes that cause neglectable burden to the wearing comfortability.

Furthermore, the three ImmHR sensors have been co-incubated in the presence of 71.4 fg/L SARS-CoV-2 NP, H1N1 HA, and RSV FP aerosols, respectively (Fig. 4e). All of the ImmHR sensor demonstrate specificity towards targeted antigen, while resonant response induced by non-specific antigen is similar to that incubated in blank aerosol samples. In addition, biochemical interferences such as ion strength, matrix composition, and pH showed little impact on the specific virus antigen detection, taking H1N1 sensor for example (Fig. 4f). Particularly, the consistent response in spiked aerosols generated from artificial saliva indicates that complex composition of saliva matrix did not hamper the specific response of antigen stimuli. Such resistance to ion strength and pH may be a major advantage over some other ultrasensitive immuno biosensors where efficient signal transduction faces a Debye-length limitation in ionic solution[8,36,37].

Prior to the stability analysis, the mass ratio of the immuno responsive hydrogel has been evaluated, as hydrogel volatilization could shrink the crosslinked network and weaken the resonant signals. After 48 h storage, the hydrogels preserved in higher relative humidity (RH) witnessed less mass loss (Supplementary Fig. 25). Thus, the prepared ImmHR sensors were preserved in 98% RH and 4 °C to sustain the detection capability. Particularly, after 5 days-storage, the hydrogel-based devices sustained 89.1% of their initial detection

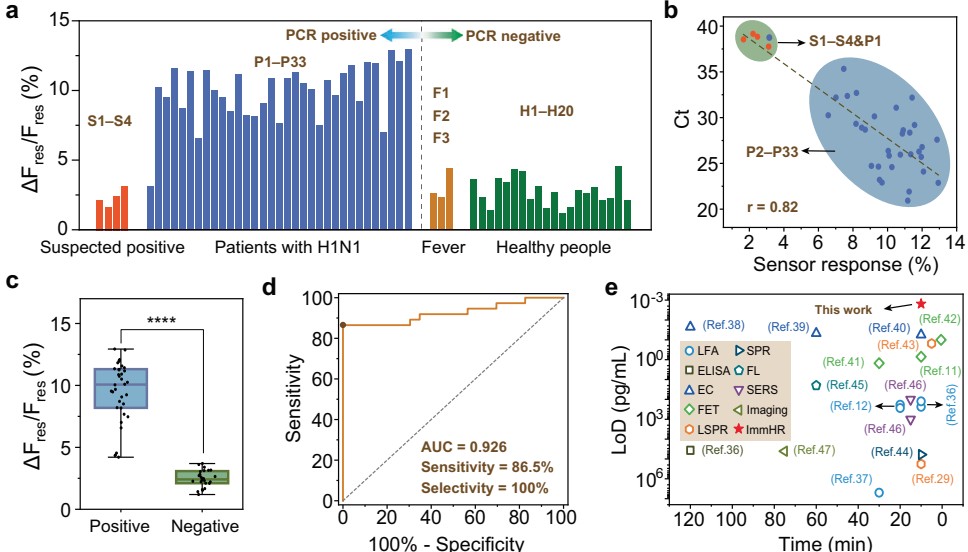

**Fig. 5 | Detection of influenza A H1N1 in clinical samples. a** Sensor response of clinical samples from suspected positive patients (S1 - S4), H1N1 positive patients (P1 - P33), fever patients yet H1N1 negative (F1-F3), and healthy volunteers (H1 - H20), with qRT-PCR as reference. **b** Correlation of sensor response and Ct values of qRT-PCR positive samples. **c** Box plot of the sensor response to H1N1 positive and negative samples ($n = 60$). Central lines indicate the median value, box limits represent the upper and lower quartiles, and whiskers indicate 1.5 × the interquartile range above and below the upper and lower quartiles, respectively. ****$P < 0.0001$, two-tailed Student's $t$-test. **d** The receiver operator characteristic (ROC) curve of the wireless immunoassay for H1N1 detection. **e** Comparison of the wireless immunoassay (this work) with other immuno sensors based on LFA, ELISA, EC, FET, LSPR, SPR, FL, SERS, and imaging.

performance, which declined to 61.7% after 7 days-storage. Thus, the period of validity is within 7 days with performance cutoff of 60% compared to the initial values under 98% RH and 4 °C (Supplementary Fig. 26). Nevertheless, similar to other hydrogel-based devices, the stability issue is a major challenge when it comes to scalable deployment of this wireless immunoassay device. It might be improved by the chemical modification of polymers to achieve low volatility gels[14], and careful tuning of the crosslink strength to sustain hydrogel's sensing performances[17].

**Clinical sample detection via wireless immunoassays**

Detection of clinical saliva aerosols faces with issues of sampling variations, target degradation during preservation and transportation, and natural fluctuations of virus load that can be greatly varied from person to person. The abundant proteins, nucleic acids, metabolites, as well as high ion strength contained in saliva further makes it challenging to achieve accurate and specific detection of targeted biomarkers. Therefore, conventional diagnostic tools generally require specialized reagent (e.g., viral transport medium) for sample preservation, complicated and time-consuming pretreatment process (e.g., virus lysis and RNA amplification) to enrich targets before eventual detection. Commercialized point-of-care testing methods such as lateral flow assay (LFA), however, suffers from disadvantages like low sensitivity and false positive/negative.

To assess the feasibility of our ImmHR-based wireless immunoassay technology for clinical sample detection, we collected nasopharyngeal swabs and saliva samples from 40 patients and 20 healthy volunteers. Specifically, 33 patients were qRT–PCR-positive for H1N1 (P1-P33) with cycle threshold (Ct) values ranging from 20–38 (Supplementary Table 1), 4 patients were suspected H1N1 positive with Ct values of 38-40 (S1-S4), 3 patients with fever but were PCR-negative (F1-F3), and 20 samples from healthy volunteers (H1-H20) were PCR negative (Fig. 5a). As demonstrated, saliva aerosols from H1N1 positive patients can be directly detected by the ImmHR sensors with resonant responses from 6.6% to 12.9% (P2-P33), compared with that of 1.2%-4.3% from healthy people (Supplementary Figs. 27, 28). Patients diagnosed with fever yet not H1N1 infection displayed 2.3%-4.4%

response by the ImmHR sensors, similar to that of healthy ones (Supplementary Fig. 29).

Samples marked as S1-S4 demonstrated Ct values of 39.15, 38.54, 38.83, and 38.74, respectively, accompanied P1 with very marginal Ct value of 37.75, which are distinct from other positive samples with Ct values of about 20-35 (Fig. 5b). Since the Ct value reflects the initial concentration of viral RNA, it can be used to characterize the viral load of collected samples. Thus, the sensor response demonstrated a good correlation with qRT–PCR Ct values with Pearson's r of 0.82, indicating that the wireless immunoassay technology can achieve quantitative virus detection. The statistic results further convinced that H1N1 positive samples are clearly distinguishable compared with negative ones ($P < 0.0001$), demonstrating the potential of this method for diagnosis of virus infection (Fig. 5c). In addition, the receiver operator characteristic curve of the ImmHR sensor shows high accuracy for H1N1 detection (area under the curve (AUC) = 0.926), with 86.5% positive predictive agreement (sensitivity) and 100% negative predictive agreement (selectivity) relative to the RT–qPCR results (Fig. 5d).

By contrast, the commercial LFA strip widely used for fast H1N1 immunoassays (10-15 min) barely demonstrated distinguishable colorimetric response until the antigen concentration reached to 100 ng/mL (Supplementary Fig. 30a). In clinical validation, it only detected 5 true positive and 21 true negative samples out of 37 PCR positive and 23 PCR negative ones, respectively (Supplementary Fig. 30b). Compared with recent progress reported on fast and sensitive immunoassays (Fig. 5e), including LFA[12,38,39], enzyme linked immunosorbent assay (ELISA)[38], electrochemical sensors (EC)[40–42], field-effect transistors (FET)[11,43,44], localized surface plasmon resonance (LSPR)[29,45], surface plasmon resonance (SPR)[46], fluorescence (FL)[47], surface enhanced Raman scattering (SERS)[48], and virus particle imaging[49], ImmHR sensor-based wireless immunoassay is featured with combined advantages of rapid detection (<10 min), high sensitivity (LoD down to fg/L or sub-fg/L), facile device configuration, and compatibility with wearable integration. Nucleic acid detection, such as PCR or clustered regularly interspaced short palindromic repeats (CRISPR) may demonstrate superior sensitivity and accuracy

(Supplementary Fig. 31 and Supplementary Table 2), however, the hours of amplification or complicated RNA extraction hinder their application for fast and facile virus surveillance.

## Discussion

Compared with nucleic acid testing, immunoassays have been considered more of a complementary approach to give a fast yet preliminary results (usually qualitative and less accurate). Nevertheless, recent endeavors present substantial progress in immunoassays through tailoring nanomaterials/nanostructures and developing efficient transducing techniques to pursue fast and sensitive detection of virus antigens or antibodies. However, a comprehensive solution to challenges including sensitivity, programmability, specificity, robustness to complex biological matrix, as well as on-site detection remains to be explored. The ImmHR-based wireless immunoassay technology is proposed to address these challenges through tuning of responsive hydrogel network and amplifing transduction of its mechanical response. Compared with conventional immuno sensors, the ImmHR sensors offer advantages with respect to immuno sensing mechanism and signal readout scheme.

First, although bio-responsive hydrogels have been proposed to capture and quantify biomolecules such as glucose[32], nucleic acids[17], proteins[50], and even virus particles[51], most of these hydrogels are slow to response and provide limited sensitivity and LoDs for clinical diagnosis. Quite different from previous works that immuno crosslinks are uniformly dispersed in the hydrogel network[19,52,53], the ImmHR sensor conjugates antibodies on AuNPs thus immuno crosslinks are focused around sites where AuNPs are immobilized. Due to the steric hindrance of AuNPs, dense AAm network cannot form locally, thus the AuNPs are immobilized within the polymer network solely by the immuno crosslinks, leading to the concentration of strain energy. The semi-stable immobilization of AuNPs can be disturbed by specific virus stimuli (competitive antigen binding) that break the immuno crosslinks and simultaneously trigger a rapid and intense release of the accumulated strain energy. In addition, the incorporation of AuNPs also results in a much lower storage modulus of the hydrogel, indicating a more elastic network. Macroscopically, both of these factors contribute to amplified hydrogel swelling to be transduced by the coupled resonators. Such mechanism also endows highly specific hydrogel response as the binding sites of AuNPs-Ab are pre-occupied by the immuno crosslinks, therefore interferences from other biomolecules are blocked. On these basis, rapid and ultrasensitive immunoassay has been realized. Validation in clinical samples indicates the high accuracy (AUC = 0.926) of this method for on-site respiratory virus surveillance.

Second, the incorporation of ImmHR sensors and RF readout network provides a wireless, multiplexed, and programmable immunoassay platform. Inspired by recent advances that integrate chipless RF sensors (inductance coils, capacitors, split ring resonators) with flexible manufacturing techniques[23,26–28], we further endow sensitive and specific immunoassay capability to the wireless sensing technology. The configuration of miniaturized resonators and flexible IR coils enables robust resonant frequency readout regardless of deformation or read coil misalignment. The programmability of resonant frequency by varying the geometry and interlayer thickness of individual ImmHR sensors enable multiplexed sensor readout by a paralleled IR network. Due to the neglectable crosstalk among different sensor channels, we realized simultaneous detection of four separate resonant peaks that corresponds to three ImmHR sensors (for SARS-CoV-2, H1N1, and RSV detection, respectively) and a wireless temperature sensor. However, with improved quality factor of the ImmHR sensors, more sensors can be accommodated in the multiplexed detection as sharper resonant peaks are less likely to interfere with adjacent ones. As a demonstration, the ImmHR sensors and IR coils can be integrated in a face mask as a RFID device for on-site and noninvasive surveillance of respiratory

virus infection. Notably, this wireless immunoassay technology is automatic and passive, which merely needs a single scanning by approaching the sensor-end (for instance, a face mask) to the readout-end (portable and inexpensive VNA with a read coil). Such easy and fast operation facilitate miniaturization and wearable integration for on-site virus protein detection, compared with other ultrasensitive immunoassay methods which are time-consuming[7,10] or need skilled operation[8,11].

In summary, we demonstrated a wireless immunoassay technology for detection of SARS-CoV-2, H1N1, and RSV virus antigens in aerosols by simple substitution of antigen/antibody recognition elements, which all showed fast response (<10 min) and fg/L or sub-fg/L LoDs. Thus, the wireless immunoassay technology can not only work as a versatile immuno sensing platform to capture and quantify virus protein biomarkers, but also detect other crucial yet low-abundance protein biomarkers in varied biofluids (inflammatory biomarkers in sweat[54], cytokines in wound exudate or interstitial fluid[43,55,56], and exosomes in blood plasma[22,57], etc.). In addition, critical tuning of bio-responsive hydrogels from a rich repertoire of polymers and bio-recognition elements (aptamers[21], DNA[17], or CRISPR tools[58,59]) could include metabolites and nucleic acids within the detectable analytes as well. Furthermore, the versatility to combine different types of bio-recognition elements within a set of ImmHR sensors, or within one single sensor with rationalize spatial distribution is expected to substantially improve the accuracy of biomarker detection and disease diagnosis (infectious diseases and cancers, etc.). Nevertheless, similar to other hydrogel-based devices, the stability issue is a major challenge when it comes to scalable deployment of this wireless immunoassay device. It might be improved by the chemical modification of polymers to achieve low volatility gels[14], and careful tuning of the crosslink strength to sustain hydrogel's sensing performances[17].

## Methods

### Hydrogel formation and characterization
**Synthesis of vinyl-antigen.** The synthesis of vinyl-antigen is adapted from method of previous report[19]. NSA (Sigma-Aldrich) was added to the sterile water (Solarbio Co., Ltd) containing the antigens (SARS-CoV-2 NP antigen, 40588-V08B; influenza A H1N1 HA antigen, 11684-V08B; human RSV A2 FP, 11049-V08B; all purchased from Sino Biological), with NSA/antigen molar ratio of 15:1, the reaction was then incubated at 36 °C for an hour to introduce vinyl groups into the antigen. The resultant vinyl-antigen was separated from the unreacted NSA with the centrifugal filter (Merck Millipore) of 10-kDa molecular weight cutoff by centrifuging at 14000 g at 4 °C for 15 min. The purified vinyl-antigens were subsequently concentrated to 0.05 mg/mL. The modification of vinyl-antibody for preparation of antibody-crosslinked hydrogel follows the same procedure.

**Determination of $K_D$ values between antigen-antibody.** The $K_D$ values of antigen-antibody binding is determined by the SPR system (SPRm 200, Biosensing Instrument, USA). First, gold chips were ultrasonic bathed in ethanol and DI water for 10 min, respectively, and dried by nitrogen flow. Then the cleaned gold chips were soaked in 0.01% Poly-L-lysine (Sigma-Aldrich) for 10 min to form a self-assemble layer for antibody immobilization, and rinsed with DI water followed by nitrogen drying. Then gold chips were soaked in 1 × PBS containing 10 μg/mL SARS-CoV-2 NP antibody (rabbit mAb, 40143-R001, Sino Biological) at room temperature for 150 min, and rinsed with 1 × PBS. Then 1 mL of 1 μg/mL pristine SARS-CoV-2 NP antigen and corresponding vinyl-antigen were flowed to the gold chips at a rate of 150 μL/min, respectively, followed by the 1 × PBS rinse. The dynamic association and dissociation kinetics were measured by the SPR system, and the $K_D$ values were obtained from the fitting model of the SPR intensity signals (Supplementary Fig. 2).

**Synthesis and characterization of AuNPs-Ab.** The AuNPs were synthesized by the trisodium citrate reduction method. In brief, 100 mL of 1 mM chloroauric acid ($HAuCl_4$, Sigma-Aldrich) was heated to boiling, and 10 mL of 38.8 mM trisodium citrate ($Na_3C_6H_5O_7$, Sigma-Aldrich) was immediately added. The mixture was boiled for another 20 min and cooled down, centrifuged at 10000 g for 15 min to concentrate to 2 × AuNPs solution. Then 16 ng of SARS-CoV-2 NP antibody, H1N1 HA antibody (rabbit mAb, 11684-MM05), and RSV FP antibody (rabbit mAb, 11049-R338) were added to 1 mL of 2 × AuNPs and incubated overnight at 4 °C, respectively. All the antibodies were purchased from Sino Biological. The morphology and size distribution of as-prepared AuNPs, AuNPs-Ab, and aggregated AuNPs-Ab were characterized by transmission electron microscope (H-9500, Hitachi, Japan) and Nano analyzer (ZEN3600, Malvern, UK), respectively.

**Copolymerization and crosslinking.** The synthesis of AuNPs-Ab crosslinked hydrogels follows a two-step gelation strategy under room temperature (-25 °C). In the first step, 30 mg AAm (Sigma-Aldrich), 0.8 μg of vinyl-antigen, 3 μL of 0.8 M TEMED (Sigma-Aldrich) and 1 μL of 2 M APS (Sigma-Aldrich) were added into 124 μL of DI water. The mixture was incubated at room temperature for about 1 min to enable copolymerization of AAm and vinyl-antigen. In the second step, 2 μL of 20 mg/mL MBAA (Sigma-Aldrich) and proper amount of AuNPs-Ab were added into the mixture to initiate covalent crosslinking and immuno crosslinking, respectively. The crosslinking strength was adjusted by the concentration of AAm and AuNPs-Ab, respectively. The hydrogel precursor generally reaches gelation within 5 min. Preparation of PAAm hydrogel follows the same procedure in the absence of AuNPs-Ab. The antibody crosslinked hydrogels, on the other hand, substitutes the AuNPs-Ab by vinyl-antibody, while other compositions unchanged. All the hydrogels were soaked in DI water for 15-30 min to remove the unreacted reagents.

**Characterization.** The hydrogels were frozen dried and observed by scanning electron microscope (GEMINI 300, Zeiss, Germany). The size distribution of micropore and nanopore were analyzed by ImageJ software. The Au and N element mapping of aggregated AuNPs-Ab was acquired by energy dispersive spectroscopy under 12 keV (Bruker Nano, Bruker, Germany). The rheological properties of hydrogels were tested by a rotational rheometer (RS6000, HAKE, US) at 25 °C, the G' and G" were measured using a shear stress sweep test ranging from 0.7−23.4 Pa.

**Hydrogel swelling tests.** To characterize the swelling ratio of the hydrogels, the hydrogel precursor was casted into a well with 8 mm diameter and 2 mm height. The hydrogels were immersed in DI water and 1 ng/mL antigen for 48 h, respectively, and measured the volume change by film ruler (Supplementary Fig. 5c). Since the hydrogel is isotropic in composition and responsiveness, the swelling ratio is calculated by equation:

$$\text{Swelling ratio} (\%) = \frac{r^3 - r_0^3}{r_0^3} \qquad (1)$$

where $r$ is the diameter of swelled hydrogels and $r_0$ is the diameter of initial hydrogel.

## Numerical simulation of the RF resonators
CST Studio Suite 2022 has been used for electromagnetic simulation. The software runs under Windows 10, and the computer hardware for simulation is Lenovo ThinkBook 16 with i7 Inter core, 16 GB RAM. Return loss and resonant frequencies of the ImmHR sensors were simulated from 0−2 GHz using the finite-difference time-domain. The geometry of the resonators, IR coil, and read coil is defined by the 3D modeling (Supplementary Fig. 7). The read coil is a circular one-port

ungrounded antenna. The hydrogel dielectric interlayer is represented by water ($\varepsilon_r = 81$). The conducting plates of IR coil and read coil are lossy copper with thickness of 100 μm, while the resonators are gold. The substrate of IR coil is lossy polyimide (PI, $\varepsilon_r = 3.5$) while the substrate of read coil is FR-4 ($\varepsilon_r = 4.3$). The distances of sensor to IR and IR to read coil are both 0.2 mm.

For split ring resonators (SRR), the capacitance of two reversely aligned SRR can be approximately expressed as:

$$C = \frac{\varepsilon_r}{d} \left[ w(b+c) - 2w^2 - 2hw \right] \qquad (2)$$

where C is the equivalent capacitance, $\varepsilon_r$ is the relative permittivity of hydrogel dielectric interlayer, d is the thickness of the hydrogel, w denotes the metal width, h denotes the split width, b and c are the length and width of the split ring (Supplementary Fig. 16a). Besides, the resonant frequency of the ImmHR sensor can be expressed as:

$$F_{res} = \frac{1}{2\pi\sqrt{LC}} \qquad (3)$$

where, $F_{res}$ is the resonant frequency, L and C are the equivalent capacitance and inductance, respectively. Thus, manipulation of initial hydrogel thickness and split ring geometry can tune the equivalent capacitance of the ImmHR sensors, and subsequently tune their $F_{res}$ for simultaneous readout of multiple resonant peaks in the RF spectrum. Similarly, the $F_{res}$ of spiral coils can be tuned by their dielectric interlayer thickness and geometry[25].

## Fabrication of resonator, IR, and read coil
To fabricate the resonators, nickel, copper, and gold foils are coated on 160 μm thick polyethylene terephthalate (PET) substrate with thickness of 3, 12, and 1 μm, respectively (Supplementary Fig. 6b, c). The substrates and metal foils are patterned to predefined geometry by laser-cutting. Then resonators are rinsed by ethanol and DI water, respectively, dried by nitrogen and stored for use. The IR coils are fabricated on 25 μm thick PI substrate with 35 μm thick copper foil by flexible printed circuit board (FPCB) technique. The circular read coil (copper, 1 Oz) is fabricated on PCB board (FR-4, 1.6 mm thickness), with a coil width of 4.5 mm. The typical geometry of IR and read coil are displayed in Supplementary Fig. 16b.

## Assembly of ImmHR sensors
As illustrated in Supplementary Fig. 6a, the as-fabricated resonators are spin-coated by SU-8 (Sigma-Aldrich) to form an insulation layer to protect the gold surface. Then the resonators are treated by oxygen plasma (TS-PL02, Dongxin Instrument, China) for 3 min to improve the hydrophilicity of the resonators for robust adhesion of hydrogels. After plasma cleaning, the bottom resonators are placed in the wells of an Ecoflex (BASF SE) mold, and proper amount of hydrogel precursor is quickly filled into the wells, followed by the capping of reversely aligned top resonators. The thickness of the hydrogel is controlled and evaluated by the volume of precursor added. After 5 min, the assembled ImmHR sensors are demolded from the flexible Ecoflex mold, and soaked in 1 × PBS for 15-30 min to remove the unreacted reagents.

## Fabrication of wireless temperature sensor
The fabrication of wireless temperature sensor is similar to that of ImmHR sensors. Nevertheless, polyethylene glycol (PEG-1500, Sigma-Aldrich) is adopted as the temperature responsive interlayer[28]. In brief, 1 g/mL PEG-1500 was prepared in DI water and dissolved by heating in 70 °C water bath. Then PEG-1500 was casted on 2.25 turn spiral coils and quickly capped by reversely aligned top coils. Finally, the assembled sensors were encapsulated by in Ecoflex and heated under 70 °C until they are cured. Since the $\varepsilon_r$ of PEG-1500 is a function of

temperature[34], rise of breath temperature can increase the equivalent capacitance of the wireless temperature sensor, thus reducing the resonant frequency for potential fever monitoring.

## Generation of antigen spiked aerosols

To prepare antigen spiked aerosols, gradient concentrations of antigen (1 μg/mL–0.1 fg/mL) were spiked in 1 × PBS or artificial saliva (R41109, Yanye Bio-Technology Co., Ltd), prepared by serial dilution. Then 3-4 mL of the antigen spiked solution was added into the liquid chamber of a medical-grade nebulizer (NE-C28, OMRON, China), which was atomized by compressed air and delivered to the testing chamber (Supplementary Fig. 14). The nebulizer generates aerosol particles with size below 5 μm, which is similar to that naturally produced by human respiration[3]. The flow rate ($f_r$) of the nebulizer is 3.5 L /min, which can atomize 2.5 mL antigen spiked solution in 10 min, thus there is a conversion factor of 0.0714 from pg/mL in solution to pg/L in aerosols, calculated as:

$$\text{Conversion factor} = \frac{C_{solution}V_{solution}}{f_r t} = \frac{1\,\text{pg/mL} \times 2.5\,\text{mL}}{3.5\,\text{L/min} \times 10\,\text{min}} = 0.0714\,\text{pg/L} \tag{4}$$

## Colorimetric assay

The as-prepared hydrogels demonstrate light blue, which gradually fades out due to AuNPs-Ab aggregation in the presence of antigen or virus stimuli. Thus, the colorimetric assay has been used to evaluate the responsiveness of the hydrogels to optimize the composition. Images of hydrogels before/after reaction were obtained by a camera (PowerShot G7 X Mark II, Canon, Japan). The colorimetric assay was performed by the ImageJ software, while the RGB parameters of each image were calibrated with the white background, respectively, to exclude environment light interferences. The colorimetric response (ΔR) is defined as

$$\Delta R = R - R_0 \tag{5}$$

where $R_0$ and $R$ are the intensity of red parameter before/after reaction, respectively. The R parameter is selected as it shows lowest relative standard deviation (Supplementary Fig. 10).

## Wireless immunoassay

For wireless immunoassay, the ImmHR sensors are fixed on one ends of the IR coils while the other ends are coupled to the read coil. The read coil is connected to a portable vector network analyzer (VNA3000) via SMA connection (Supplementary Fig. 16c). The probing of return loss is operated from 1–1000 MHz in the reflection mode. The resonant response is defined as:

$$\text{Resonant response}\,(\%) = \frac{\Delta F_{res}}{F_{res}} \tag{6}$$

where $\Delta F_{res}$ denotes the resonant frequency shift, while $F_{res}$ is the resonant frequency of the ImmHR sensor before reaction.

## Wireless temperature detection

The $F_{res}$ of the temperature sensor was calibrated by capping the sensor on water vapor from thermostatic water bath. To demonstrate its ability for real-time temperature monitoring, the temperature sensor was balanced in room temperature (25 °C), followed by exposing to human breath (about 37 °C). During this period, the return loss was probed every 5 s to plot the real-time response.

## Displacement and deformation tests

To simulate the displacement between the sensor/IR and read coil during wireless detection, the read coil was placed 2 mm away from the center of the sensor/IR (Supplementary Fig. 15). The bending tests were performed by sticking the IR on curved PET tubes with 40-80 mm bending radii. The twisting tests were performed by fixing one end of the IR on plane surface and twisting the other end by 90° with a tweezers. For all the displacement and deformation tests with IRs, the ImmHR sensors were fixed on IRs with minimal mechanical disturbance.

## Wearable integration and RF spectral analysis

The ImmHR sensors were fixed on flexible IRs and stuck on the inner surface of a KN95 face mask. For simultaneous readout of multiple sensors, the IR coils are aligned and fixed at one ends, which were parallelly and wirelessly coupled to the read coil. The sensor ends were separate from each other at least 5 mm to minimize influence of mutual inductance between adjacent sensors. Four sensors configured at 197.6, 347.8, 453.5, 622.3 MHz were accommodated into the parallel RF network. The RF spectral analysis was performed at room temperature, and ImmHR sensors were added one by one to investigate the sensor crosstalk. In addition, the sensor array was exposed to 40 °C water vapor from thermostatic water bath to study the influence of temperature on ImmHR sensors.

## Specificity and biochemical interferences

The specificity of ImmHR sensors were tested in 0.0714 ng/L SARS-CoV-2 NP, H1N1 HA, and RSV FP antigen aerosol, respectively, with blank 1 × PBS aerosol as reference. Influence of ion strength was tested in aerosols from spiked DI water, 0.1 × PBS (0.001 M), and 1 × PBS (0.01 M), while artificial saliva aerosol (pH = 6.8, containing ptyalin and mucoprotein) provided a biological matrix similar to human saliva. Influence of aerosol pH was studied by adjusting pH of DI water.

## Preservation and stability tests

The preservation humidity is controlled by the top air of saturated saline solution (Na$_2$HPO$_4$, RH = 98%; KCl, RH = 85%, NaCl, RH = 76%). The mass of hydrogels were measured every 12 h, normalized to initial value to calculate the mass ratio. The stability was tested for 7 days in the presence of 71.4 pg/L antigen aerosols.

## Clinical sample collection and sensor validation

De-identified clinical samples were collected from health individuals or individuals confirmed or suspected with influenza infection, collected under the study approved by Ethics Committee of Department of Biomedical Engineering, Zhejiang University ([2022]-8). This research complies with all relevant ethical regulations. Each participant was taken 1-2 mL saliva sample and a nasopharyngeal swab, within which the nasopharyngeal swab was tested in LFA assay, about 1 mL saliva was stored in viral transport medium and delivered for qRT-PCR test, and about 1 mL diluted saliva was atomized into aerosols directly tested by the ImmHR sensors. The exhaust aerosols were collected with other unused samples to be denatured by 75% v/v ethanol and UV irradiation for 30 min. For all the 60 participants, 37 were diagnosed with H1N1 positive (including 4 patients with marginal Ct values of 38-40), 23 were H1N1 negative (including 3 patients with fever and diagnosed with bacterial infection). Informed consents were received from all the participants.

## qRT–PCR assay of clinical samples

About 1 mL saliva samples of each participant were delivered for standard qRT-PCR tests (Matridx Biotechnology Co., Ltd). In brief, 200 μL saliva was mixed with PBS and nucleic acid extraction reagent following the manufacturer's instructions. qRT-PCR assays were performed in a 25 μL reaction system on a real-time PCR system (ABI 7500,

Thermo Fisher) as follows: 50 °C 10 min, 95 °C 5 min, 45 cycles of 95 °C 15 s, 60 °C 30 s. All the samples were detected in triplicate.

### LFA assay of H1N1 HA protein and clinical samples
For LFA assay of clinical samples, nasopharyngeal swabs were collected and soaked in extraction tubes for 30 s and pressed for 5 times following the manufacturer's instructions. Then 100 µL of the solution was dropped on the test spots of LFA strips. Images of the testing and control lines were obtained by a camera after 10 min incubation. For reference, H1N1 HA spiked artificial saliva with concentration of 1 fg/mL-1 µg/mL was tested by the LFA strips, too (Supplementary Fig. 30).

### Statistical analysis
All the experiments were performed in at least triplicate unless otherwise mentioned. The data is displayed as mean ± standard deviation ($\sigma$). For calibration plots demonstrate linear dependence against logarithmic concentration, linear fitting was performed with linear regression to determine the goodness of fit ($R^2$). For calibration plots demonstrate S-Curve dependence against logarithmic concentration, logistic fitting was performed with $R^2$ analysis. For equilibrium response, ExpAssoc fitting model was adopted. All fittings were performed by Origin (Origin 2023b Learning edition). The LoDs were calculated from the equivalent response of mean response plus 3 $\sigma$ in negative control, using the established calibration curves. All figures were edited in Adobe Illustrator 2019.

Significance test between clinical positive and negative samples was performed via a two-tailed Student's $t$-test. Correlation between Ct values and sensor response was evaluated by Pearson' r. ROC curve analysis of the clinical detection was performed to assess the sensitivity, specificity and accuracy of the wireless immunoassay technology.

### Reporting summary
Further information on research design is available in the Nature Portfolio Reporting Summary linked to this article.

## Data availability
All data supporting the findings of this study are available within the article and its supplementary files. Any additional requests for information can be directed to, and will be fulfilled by, the corresponding authors. Source data are provided with this paper.

## Code availability
No customized code used.

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

## Acknowledgements

The authors thank Shangyu People's Hospital of Shaoxing City for assistance with clinical sample collection and Hangzhou Matridx Biotechnology Co., Ltd for PCR tests. Prof. Q.L. thanks the funding support of Zhejiang Provincial Natural Science Foundation of China (No. LZ23C100001) and National Natural Science Foundation of China (Grant No. 81971703). Dr. X.L. thanks the funding support of China Postdoctoral Science Foundation (No.2023M743054).

## Author contributions

X.L. and Q.L. conceived the idea. X.L., R.S. and J.P. performed the experiments. X.L. and R.S. processed the data. X.L. drew the figures and prepared the manuscript. X.L., Z.S., Z.A., C.D. designed the resonant sensor. J.L., G.L. and H.L. assisted the optimization of the hydrogel. J.L., Y.L. and F.Z. assisted the design of clinical tests and recruitment of volunteers. All authors discussed the experimental data and results. X.L. and Q.L wrote and revised the paper.

## Competing interests

The authors declare no competing interests.
