## [Peer Review File · Nature Communications]

REVIEWER COMMENTS

Reviewer #1 (Remarks to the Author):

In this manuscript, the authors present a wireless immunoassay technology, including an immune-responsive hydrogel-modulated RF resonant sensor to capture and amplify the recognition of virus antigen. This work provides a sensitive and accurate immunoassay technology for on-site virus detection and disease diagnosis compatible with wearable integration. Even though detecting COVID-19 is no more a hot topic and the usage amount of face masks is reduced a lot. Honestly no one want to wear facemasks for a long time. However, the idea is ingenious and the results of this work are convincing, and this work can be used to detect other kind of virus. I suggest to publish this work in Nature Communications after addressing some problems.

Please see my detailed concerns.

1. In Figure 1b, the crosslink intensity between AuNPs-Ab and the 3D network was disrupted upon combining the virus with hydrogel. More detailed information is required, such as the change in mechanical properties of hydrogels.
2. In the middle picture of Supplementary Fig. 3b, the size of AuNPs-Ab exhibited three different intensity peaks. So, why? Is it possible to replace it using a new picture with a single peak, just like the AuNPs or Assembled AuNPs-Ab.
3. As a common sense, the hydrogel exhibited low stability. So how about this device?
4. Do the devices in this work need special storage conditions? And how long is the period of validity?
5. Will the device (including the sensors and the circuit) assembled with the face masks affect the air permeability and comfort?
6. In the RESULT section, the authors reported that "the hydrogels in deionized water only witnessed 37% swelling compared with that of 195% in the presence of 1 ng/mL antigen (Supplementary Fig. 4c), suggesting the sensitive volume response for signal transduction." The swelling time of 3 hours may not be enough for hydrogels. Thus, from my point of view, the conclusion is not so reliable. The authors

should improve the swelling time for the hydrogels to 7 days. Furthermore, the optical pictures of hydrogels with different volumes should be provided, indicating the swelling ratio directly.

Reviewer #2 (Remarks to the Author):

The paper tackles the important topic of wireless wearable devices for point-of-care applications. The novelty is clear and the methods are sound. However, my main concerns are related to the design of the electromagnetic interface.

1. What EM simulator was used to investigate the coil? What are the SW and HW specifications?
2. "Result" section discusses the propagation of electromagnetic field, and "Wearable integration and multiplexed readout" section states that "[...] analogue to RFID technology [...] RF link is established". However, at the reading distance of 3 mm mentioned later on, the antenna system is working inside the near-field inductive zone thanks to magnetic coupling, similarly to the NFC protocol. This ambiguity should be discussed minding the differences between the different kinds of devices, known in the literature for point of care architectures.
3. The concept in Fig. 4a could be strongly limited or outright unusable because of the effects evidenced in Supplementary Figures 14 and 16. These problems should be evidenced and discussed in-depth the "Wearable integration and multiplexed readout" section, proposing methods to overcome said limitations.
4. How much power is required to achieve a reliable RF link? Is it compatible with Specific Absorption Rate regulations?

Reviewer #3 (Remarks to the Author):

This study introduces a wireless immunoassay technology for rapid and accurate detection of respiratory virus aerosols. The technology, integrated into face masks, enables fast, on-site, and sensitive detection of virus antigens in aerosols, with a low limit of detection. Utilizing an immuno-responsive hydrogel-modulated RF resonant sensor for capturing and amplifying virus antigen recognition, the wireless immunoassay achieves simultaneous detection of three types of respiratory viruses. Direct detection of clinical samples shows high accuracy in diagnosing respiratory virus infections. This work provides a

sensitive and accurate immunoassay technology for on-site virus detection and disease diagnosis, compatible with wearable integration. The study is well-designed and comprehensive. However, some questions need to be considered and clarified before acceptance.

Line 58, please avoid using etc. in the manuscript. Same for line 110, 507, 514

Line 99, facilitate@facilitates

Line 102, do you mean infection discrimination among these listed types of viruses or infection detection or both? Please clarify.

Line 119, Will the breathing itself (absence of pristine antigen) also cause the swelling of the hydrogel to some extent? have you compared the swelling caused by the competitive binding of pristine antigen? How different are they?

Line 120, is that aggregation occurred inside the preformed hydrogel? And why the competitive antigen binding leads to severe AuNP aggregation? Is this triggered by the amount of the virus/spike proteins on the surface? Please clarify.

Line 132, when you refer to antigen stimuli, is it carried out by immersing in a solution or through an actual breathing test? And the stimuli for how long? Minutes or hours? Please clarify.

Line 148: Scar bar?? Do you mean scale bar? Same for 150 and other places in the manuscript

Line 157: "Infection symptom monitoring" is quite broad. Do you mean monitor the body temperature? Need to be very specific about what you are trying to monitor.

Line 182: "which also leads to AuNPs aggregation" is this aggregation caused by the competitive immunobinding? Or swelling? Or both? This question also reflects my previous comments on has the author carried out the control experiment of the swelling test of the hydrogel itself without the virus treatment?

Line 183: "the formation of immuno responsive hydrogel needs precise control". Can the author please clarify how could this be achieved? Through reaction time? Temperature? The author may have specified this in figure 2a but also need to be point out in the main content.

Line 216: "Although the red, green and blue216 parameters show similar trend in colorimetric assays" can you please define what are the red, green and blue parameters? If this is related to the aggregation state of the AuNPs then this needs to be clarified. I realise some of this information went into the method section from line 650, please consider to move some of those to the main text to eliminate confusion.

Line 269: "ImmHR sensors with 700 μm thick hydrogel are determined for reliable269

immunoassays in following study" have you considered the drying effect to the immune response and the resonant response of a 700um thick hydrogel? Or the hydrogel must be maintained moisturised during the whole process? Please specify this.

line 568: please change "frozen dried" to freeze-dried

Response to comments of Reviewers

Reviewer #1 (Remarks to the Author):

In this manuscript, the authors present a wireless immunoassay technology, including an immune-responsive hydrogel-modulated RF resonant sensor to capture and amplify the recognition of virus antigen. This work provides a sensitive and accurate immunoassay technology for on-site virus detection and disease diagnosis compatible with wearable integration. Even though detecting COVID-19 is no more a hot topic and the usage amount of face masks is reduced a lot. Honestly no one want to wear facemasks for a long time. However, the idea is ingenious and the results of this work are convincing, and this work can be used to detect other kind of virus. I suggest to publish this work in Nature Communications after addressing some problems.

Please see my detailed concerns.

Agreed.

Thank the reviewer for your interest in our wireless immunoassay system for virus detection and the positive comments on the novelty and results of our work. As the reviewer suggests, the use of a facemask is one of the demonstrations, yet the designed sensing strategy and devices can be used for surveillance of various kind of virus, which adds glamour to this work. We have carefully addressed the concerns raised by the reviewer and revised our manuscript accordingly.

1. In Figure 1b, the crosslink intensity between AuNPs-Ab and the 3D network was disrupted upon combining the virus with hydrogel. More detailed information is required, such as the change in mechanical properties of hydrogels.

Agreed.

Thank the reviewer for this insightful comment. As the reviewer suggests, the mechanical properties, such as the shear storage modulus (G') can be used to characterize the change of crosslink intensity disrupted upon combining the virus with hydrogel. We have conducted the rheological tests of the hydrogel before/after immunoassay (Figure R1), which showed that the G' of hydrogel decreased from about 78 Pa to 26 Pa after competitive immuno binding, indicating that the 3D network has been disrupted. Such phenomenon is expected as the breaking of immuno crosslinks decrease the overall crosslink intensity, and hydrogels with lower crosslink intensity demonstrate lower shear storage modulus.

Figure R1 Shear storage modulus of hydrogels before/after competitive immuno binding.

On Page 5 of Supplementary information, Figure R1 has been added as revised Supplementary Fig. 3, and the caption has been added as “Mechanical property of hydrogels before/after competitive immuno binding.”

On Page 5, Line 118-121 of the manuscript, we have revised the text as “Thus, competitive binding of pristine antigen (virus stimuli) can break these immuno crosslinks and disrupt the preformed hydrogel network, as indicated by the lower storage modulus after immunoassay (Supplementary Fig.3).”

2. In the middle picture of Supplementary Fig. 3b, the size of AuNPs-Ab exhibited three different intensity peaks. So, why? Is it possible to replace it using a new picture with a single peak, just like the AuNPs or Assembled AuNPs-Ab.

Agreed.

Thank the reviewer for the careful check. The synthesized AuNPs-Ab exhibited three different intensity peaks possibly due to the inadequate stirring and dispersion of the solution when it is tested by Nano analyzer. We have reprepared the AuNPs-Ab solution and tested its size distribution, as shown in Figure R2, it exhibited only one distinct peak.

Figure R2 Size distribution of AuNPs-Ab.

We have replaced the middle picture of Supplementary Fig. 4b with Figure R2 on Page 6, Supplementary information, as the reviewer suggested.

Supplementary Fig. 4b Size distribution of AuNPs, AuNPs-Ab, and AuNPs-Ab aggregated by 1 ng/mL antigen in solution, respectively.

3. As a common sense, the hydrogel exhibited low stability. So how about this device?
Agreed.

Thank the reviewer for this insightful comment. As the reviewer points out, hydrogel sensors generally exhibited low stability. As depicted in Supplementary Fig. 26, the hydrogel-based devices proposed in this work can attain 89.1% of the detection performance compared to the as-prepared ones. However, obvious decline of sensing performance has been observed after a week. We attribute it to the slow dehydration effect (Supplementary Fig. 25) of the gel and the inactivation of antibodies despite the devices were preserved in highly humid, cold environment (98% RH, 4 °C in refrigerator).

Supplementary Fig. 25 Influence of preservation humidity on hydrogel dehydration. The RH is modulated by storing the hydrogels in the top air of saturated saline solution (Na_2HPO_4 , RH =98%; KCl, RH=85%, NaCl, RH=76%).

Supplementary Fig. 26 Stability of the ImmHR sensors in two weeks. The responses were tested with H1N1 sensors in 71.4 pg/L H1N1 antigen aerosols. The hydrogel devices were stored in 98% RH and 4 °C.

On Page 19, Line 514-578, Discussion section, we have discussed the general challenge of stability issue, added as “Nevertheless, similar to other hydrogel-based devices, the stability issue is a major challenge when it comes to scalable deployment of this wireless immunoassay device. It might be improved by the chemical modification of polymers to achieve low volatility gels,¹ and careful tuning of the crosslink strength to sustain hydrogel’s sensing performances.²”

Reference:

1. Wang, B. et al. Wearable bioelectronic masks for wireless detection of 783 respiratory infectious diseases by gaseous media. *Matter* 5, 4347-4362, (2022).
2. Zhang, K., Feng, Q., Fang, Z., Gu, L. & Bian, L. Structurally Dynamic Hydrogels for Biomedical Applications: Pursuing a Fine Balance between Macroscopic Stability and Microscopic Dynamics. *Chem Rev* 121, 11149-11193, (2021).

4. Do the devices in this work need special storage condition? And how long is the period of validity?

Agreed.

Thank the reviewer for this thoughtful comment. Followed with the comment and the response above, we do need to store our devices in 98% RH and 4 °C to prolong their stability. However, this storage condition is not very harsh for lab use or potential transportation for commercial use, as refrigerators and cold-chain storage at this humidity and temperature is easily accessible. As shown above, our hydrogel sensors sustained 89.1% of their initial detection performance after 5-days-storage, which declined to 61.7% after 7-days-storage. Therefore, if we set the validity cutoff as 60% of its initial sensing performance, the period of validity is within 7 days.

On Page 15, Line 375-378, we have modified the text as “Particularly, after 5-days-storage, the hydrogel-based devices sustained 89.1% of their initial detection performance, which declined to 61.7% after 7-days-storage. Thus, the period of validity is within 7 days with performance cutoff of 60% compared to the initial value under 98% RH and 4 °C.”

5. Will the device (including the sensors and the circuit) assembled with the face masks affect the air permeability and comfort?

Agreed.

Thank the reviewer for the insightful comment. As the reviewer suggests, the permeability of face mask devices need to be taken into account, as it influences the comfortability and wearing experiences. We have conducted air permeability tests by measuring the water volatile efficiency with conventional face mask and our face mask device capped on water-containing beakers, respectively, similar to previous reports.³ The volatile percentage indicates the air permeability. The volatile ratio was measured by the water vapor loss under 37°C after 6 hours with face masks capping on water-containing beakers with/without wireless immunoassay devices integrated, respectively. As shown in Figure R3, 15.59% water vapor volatile has been observed in face mask with the wireless immunoassay device, close to that of 16.61% in a conventional face mask. Therefore, integration of the wireless immunoassay device in a face mask demonstrates similar air permeability. Apart from the air permeability, the wireless immunoassay device is less than 1.5 mm thick and weights 10 g, thus the thin and lightweight device will not cause extra burden to the wearing comfortability of the integrated face mask. Besides, the device is properly adhered and encapsulated on the inner surface of the face mask by medical tapes, rendering comfortable wearing.

Figure R3 Air permeability of face masks with and without the wireless immunoassay devices.

On Page 26, Supplementary information, Figure R3 has been added as revised Supplementary Fig. 24, and the caption has been added as “Air permeability of face masks with and without the wireless immunoassay devices.”

On Page 13, Line 349-354, we have modified the text as “Besides, the integration of the ImmHR sensors and IR coils in a face mask demonstrates similar air permeability to those conventional face masks (Supplementary Fig. 24). Apart from the air permeability, the wireless immunoassay device is thin and lightweight (< 1.5 mm thick and 10 g weight), properly adhered and encapsulated on the inner surface of the face mask by medical tapes that cause neglectable burden to the wearing comfortability.”

Reference:

3. Ma, X. et al. A monolithically integrated in-textile wristband for wireless epidermal biosensing. *Sci Adv* 9, eadj2763, (2023)

6. In the RESULT section, the authors reported that “the hydrogels in deionized water only witnessed 37% swelling compared with that of 195% in the presence of 1 ng/mL antigen (Supplementary Fig.4c), suggesting the sensitive volume response for signal transduction.” The swelling time of 3 hours may not be enough for hydrogels. Thus, from my point of view, the conclusion is not so reliable. The authors should improve the swelling time for the hydrogels to 7 days. Furthermore, the optical pictures of hydrogels with different volumes should be provided, indicating the swelling ratio directly.

Agreed.

Thank the reviewer for the insightful comment. Since our hydrogel sensors produced distinct response for virus antigen detection within 10 minutes, previously we did not conduct equilibrium swelling test of the hydrogel. However, as the reviewer suggested, from the kinetics point of view, unveiling the equilibrium swelling would better support the conclusion of this work. As the reviewer pointed out, we have conducted the swelling test until the hydrogels reached to equilibrium swelling, and recorded the optical pictures of hydrogels and recalculated the swelling ratio. As shown in Figure R4, the hydrogels achieved equilibrium swelling after 48 hours. The hydrogels immersed in 1 ng/mL antigen exhibited faster swelling rate and higher swelling ratio compared to that immersed in deionized water, which reached to 471% and 387% after 48 hours, respectively. The faster swelling rate in 1 ng/mL antigen can be attributed to the break of immuno crosslinks that additionally decreased the overall crosslink intensity that facilitated faster water molecules absorption, which was lacking in DI water. When reaching to equilibrium swelling, the swelling ratio in 1 ng/mL antigen was also higher than that in DI water, the results corresponded well with the designed mechanism that upon stimuli from virus proteins, the hydrogel network would be disrupted more significantly and induce higher swelling ratio.

Figure R4 Equilibrium swelling ratio of the hydrogel in DI water and 1 ng/mL antigen spiked solution. Left panel shows the optical images of swelled hydrogels.

On Page 7, Supplementary information, Supplementary Fig. 5c has been replaced by Figure R4. The caption has also been revised as “(c) Equilibrium swelling ratio of the hydrogel in DI water and 1 ng/mL antigen spiked solution. Left panel shows the optical images of swelled hydrogels.”

On Page 22-23, Line 577-582, Method section, Hydrogel swelling test has been revised as “The hydrogels were immersed in DI water and 1 ng/mL antigen for 48 hours, respectively, and measured the volume change by filming ruler (Supplementary Fig.5c). Since the hydrogel is isotropic, the swelling ratio is estimated by

$$\text{Swelling ratio (\%)} = \frac{r^3 - r_0^3}{r_0^3}$$

where r is the diameter of swelled hydrogels and r_0 is the diameter of initial hydrogel.”

On Page 5, Line 138, we have revised the text as “Specifically, the hydrogels immersed in 1 ng/mL antigen exhibited faster swelling rate and higher swelling ratio compared to that immersed in deionized water, which reached equilibrium after 48 hours, respectively (Supplementary Fig. 5c). The faster swelling rate in 1 ng/mL antigen can be attributed to the break of immuno crosslinks that additionally decreased the overall crosslink intensity that facilitated faster water molecules absorption, which was lacking in DI water. When reaching to equilibrium swelling, the swelling ratio in 1 ng/mL antigen was also higher than that in DI water, the results corresponded well with the designed mechanism that upon stimuli from virus antigens, the hydrogel network would be disrupted more significantly and induce higher swelling ratio.”

Reviewer #2 (Remarks to the Author):

The paper tackles the important topic of wireless wearable devices for point-of-care applications. The novelty is clear and the methods are sound. However, my main concerns are related to the design of the electromagnetic interface.

Agreed.

We sincerely thank the reviewer for the positive comments and insightful suggestions to clarify the design of the electromagnetic interface. We have carefully addressed the concerns listed below and revised our manuscript accordingly.

1. What EM simulator was used to investigate the coil? What are the SW and HW specifications?

Agreed.

Thank the reviewer for the thoughtful comment. We used CST Studio Suite 2022 to realize the EM simulation of split ring resonators and spiral coils. The software (SW) runs under Windows 11, and the computer hardware (HW) for simulation is Lenovo ThinkBook 16 with i7 Inter core, 16 GB RAM.

As the reviewer suggests, we have clarified the EM simulator, software, and hardware specifications on Page 31, Line 584-586 as “CST Studio Suite 2022 has been used for electromagnetic simulation. The software runs under Windows 10, and the computer hardware for simulation is Lenovo ThinkBook 16 with i7 Inter core, 16 GB RAM.”

2. “Result” section discusses the propagation of electromagnetic field, and “Wearable integration and multiplexed readout” section states that “[...] analogue to RFID technology [...] RF link is established”. However, at the reading distance of 3 mm mentioned later on, the antenna system is working inside the near-field inductive zone thanks to magnetic coupling, similarly to the NFC protocol. This ambiguity should be discussed minding the differences between the different kinds of devices, known in the literature for point of care architectures.

Agreed.

Thank the reviewer for the insightful comment and kind suggestion. As the reviewer pointed out, we have been aware that our wireless immunoassay device can only work within several millimeters away from the read coil. This feature, as the reviewer suggested, makes it more like the NFC devices basing on near-field magnetic coupling, rather than RF devices with longer working distances. From our point of view earlier, NFC technique is a strictly defined technology that the device must be resonant

at 13.56 MHz according to ISO/IEC 18000-3 air interface. Therefore, we regarded it as an analogue to RF devices considering the resonant frequencies of our sensors are around 200~600 MHz. Nevertheless, after careful consideration, we agree with the reviewer that, in the concept depicted in Fig. 4a, it's more reasonable to describe this device as an analogue to NFC device to emphasize the constrain of near working distance. Since NFC technique is a particular form of RFID technique, we believe such metaphor is clearer to reduce the ambiguity.

As the reviewer suggested, we have clarified the difference between our device and conventional RF devices on Page 13, Line 333-337, "Despite that the ImmHR sensors demonstrate about 200~600 MHz resonant peaks, the integrated face mask device only works by approaching it (similar to a near field communication (NFC) card) to the reader. The wireless link between the Face Mask Card and the reader establishes within several millimeters."

On Page 14, Fig. 4a has been modified as "Face Mask RFID Card" has been revised as "Face Mask Card", and "RF link" has been revised as "Wireless link", "RF reader" has been revised as "Reader" to avoid the misunderstanding about the near working distance nature of our device.

On Page 14, Line 363, the caption has been revised as "a, Illustration of wearable aerosol detection with the ImmHR sensor array for respiratory virus surveillance. The integrated face mask device works similarly to near field communication (NFC) devices, which responds when approaches to the reader device."

Revised Fig. 4a Illustration of wearable aerosol detection with the ImmHR sensor array for respiratory virus surveillance. The integrated face mask device works similarly to near field communication (NFC) devices, which responds when approaches to the reader device.

3. The concept in Fig. 4a could be strongly limited or outright unusable because of the effects evidenced in Supplementary Figures 14 and 16. These problems should be

evidenced and discussed in-depth the Wearable integration and multiplexed readout” section, proposing methods to overcome said limitations.

Agreed.

Thank the reviewer for the insightful comment and kind suggestion. We are so sorry about the ambiguity that Supplementary Fig. 15 (original Supplementary Fig. 14) and Fig. 17 (original Supplementary Fig. 16) may seem to you. We have to clarify that, as demonstrated in Supplementary Fig. 15, the displacement interference was obvious if the sensor was directly coupled to a read coil. However, to avoid such interferences, we had incorporated an intermediate coil (IR coil), as demonstrated in Fig 3a. In this configuration, the sensor was fixed on one end of the IR coil, while the other end was coupled to the read coil, which alleviated the displacement interferences.

Supplementary Fig. 15 Influence of sensor displacement without IR coils. Left: Illustration of the origin point and direction of sensor displacement. Right: Resonant frequency shift upon sensor displacement in IR-free configuration.

Fig. 3a Misalignment test between IR and read coil, including left, right, inward, outward displacement of 2 mm, respectively. The ImmHR sensor is fixed on the IR coil by double-side tape.

The displacement interference is mainly due to the variation of parasitic capacitance during the displacement of two closely approached circuit component.^{1,2} Most recently, Dautta, M. et al. proposed that the use of IR coils, namely intermediate coils, helps stabilize the spectral readout during displacement between the readout coil and the network.⁴ The unwanted parasitic capacitance is suppressed as the sensors were fixed on IR coils. Such flexible, grounded multiple magnetic coupling has been utilized in other report.⁵ As we have demonstrated in Fig 3a, with an IR coil and sensor fixed on it, the displacement hardly influenced the resonant peaks. To summarize this point, results in Fig. 3a proved that adoption of IR coils suppressed the influence of displacement effect, while results in Supplementary Fig. 15 was a contrast to emphasize the importance of IR coils in wearable devices.

We have to clarify that, the results in Supplementary Fig. 17 actually proves that due to the adoption of IR coils, the bending of the whole network did not significantly influence the resonant frequency. Such phenomenon corresponds well with previous results that the bending of these magnetic coupling devices has little impact on the robust readout of the resonant peak signals.^{5,6}

To overcome the working distance constrain in Fig. 4a, one solution is to enhance the emission field with larger read coils and higher power. Note that the read coil we used is a homemade planar PCB coil for standard lab tests, which can be optimized by fabricating with copper spiral wires for higher inductance. The stronger field emission enables better energy harvesting by the sensor network, thus wireless readout can be realized at longer working distances, as demonstrated recently.⁷ Influence or sensor displacement, however, can be further suppressed by better grounding design, for instance, addition of another layer of aluminium foil on the back side of the IR coils.⁶

Supplementary Fig. 17 Influence of bending radii on sensor readout. Left: Resonant frequency of the ImmHR sensor on IR with bending radii of 40, 60, 80 mm, compared to flat status. Right: The bending radii of a curved face mask. The similar

resonant peaks indicate that bending of these magnetic coupling devices has little impact on the robust readout of the return loss signals.

As the reviewer suggested, we agree that it is essential to clarify the results in Supplementary Figures 15 and 17 with more in-depth discussion in the manuscript, and propose methods to overcome potential limitations.

On Page 10, Line 252, we have added the discussion as "However, direct coupling between an resonant sensor and a read coil witnesses F_{res} shift upon relative displacement (Supplementary Fig. 15). The resonant frequency shift is mainly due to the variation of parasitic capacitance during the displacement of two closely approached circuit component.^{1,2} To handle this misalignment disturbance, ImmHR sensors are fixed on flexible IR coils that magnetically couple to the read coil (Supplementary Fig. 16a-c), which helps stabilize the spectral readout between the readout coil and the network.^{3,4} The adoption of IR coils is simple to maintain due to the flexible nature of the IR. Such flexible, grounded magnetic coupling has been utilized to suppress the parasitic capacitance previously.^{5,6} Due to the use of IR coils, the impact of misalignment effect has been eliminated, resulting in a stable F_{res} readout regardless of the amplitude of return loss (Fig.3a)."

On Page 10, Line 258, we have added the discussion of "Due to the adoption of IR coils, the bending of the whole network did not significantly influence the resonant frequency. Such phenomenon corresponds well with previous results that the bending of these magnetic coupling devices has little impact on the robust readout of the return loss signals (Supplementary Fig. 17).^{5,6}"

On Page 10-11, Line 349, we have added the discussion of "One possible solution to further improve the signal readout is to enhance the emission field with larger read coils and higher power. Note that the read coil we used is a homemade planar PCB coil for standard lab tests, which can be optimized by fabricating with copper spiral wires for higher inductance. The stronger field emission enables better energy harvesting by the sensor network, thus wireless readout can be realized at longer working distances, as demonstrated recently.⁷ Influence or sensor displacement, however, can be further suppressed by better grounding design, for instance, addition of another layer of aluminium foil on the back side of the IR coils.⁶"

Furthermore, we have added the Supplementary Movie 1 to describe the concept of wireless immunoassay by approaching the Face Mask Card to Reader device for community infection surveillance, similarly to the identity authentication process based on near field communication technique.

Reference:

1. Dong, Y. et al. A “Two - Part” Resonance Circuit based Detachable Sweat Patch for Noninvasive Biochemical and Biophysical Sensing. *Adv Funct Mater* 33, 2210136, (2022)
2. Niu, S. et al. A wireless body area sensor network based on stretchable passive tags. *Nat Electron* 2, 361-368, (2019)
3. Carr, A. R. et al. Toward Mail-in-Sensors for SARS-CoV-2 Detection: Interfacing Gel Switch Resonators with Cell-Free Toehold Switches. *ACS Sens* 7, 806-815, (2022)
4. Dautta, M. et al. Programmable Multiwavelength Radio Frequency Spectrometry of Chemophysical Environments through an Adaptable Network of Flexible and Environmentally Responsive, Passive Wireless Elements. *Small Sci* 2, (2022)
5. Hajiaghajani, A. et al. Textile-integrated metamaterials for near-field multibody area networks. *Nat Electron* 4, 808-817, (2021)
6. Lin, R. et al. Wireless battery-free body sensor networks using near-field-enabled clothing. *Nat Commun* 11, 444, (2020)
7. Won, S. M., Cai, L., Gutruf, P. & Rogers, J. A. Wireless and battery-free technologies for neuroengineering. *Nat Biomed Eng* 7, 405-423, (2023)

4. How much power is required to achieve a reliable RF link? Is it compatible with Specific Absorption Rate regulations?

Agreed.

Thank the reviewer for the thoughtful comment about biosafety. Here we used a portable vector network analyzer (VNA3000) with low power consumption and a homemade planar PCB read coil for the probing of return loss. For passive devices like our face mask device, the power required to achieve a reliable RF link is only 3~5 mW,⁷ which makes it rather safe to satisfy the Specific Absorption Rate limit of 1.6 mW/g averaged over one gram of tissue in the head. Furthermore, such passive, NFC-like devices has been widely used in implantable biomedical electronics.⁷⁻¹⁰ On this basis, we believe our device is compatible with Specific Absorption Rate regulations.

As the reviewer suggested, we have added the discussion about the biosafety on Page 14, Line 335, “For passive devices like the Face Mask Card, the power required to achieve a reliable RF link is only 3~5 mW,⁷ which makes it rather safe to satisfy the Specific Absorption Rate limit of 1.6 mW/g averaged over one gram of tissue in the head.”

Reference:

7. Won, S. M., Cai, L., Gutruf, P. & Rogers, J. A. Wireless and battery-free technologies for neuroengineering. *Nat Biomed Eng* 7, 405-423, (2023)

8. Cai, X. et al. A wireless optoelectronic probe to monitor oxygenation in deep brain tissue. *Nat Photonics*, (2024)
9. Choi, Y. S. et al. A transient, closed-loop network of wireless, body-integrated devices for autonomous electrotherapy. *Science* 376, 1006-1012, (2022)
10. Zhang, H. et al. Wireless, battery-free optoelectronic systems as subdermal implants for local tissue oximetry. *Sci Adv* 5, eaaw0873, (2019)

Reviewer #3 (Remarks to the Author):

This study introduces a wireless immunoassay technology for rapid and accurate detection of respiratory virus aerosols. The technology, integrated into face masks, enables fast, on-site, and sensitive detection of virus antigens in aerosols, with a low limit of detection. Utilizing an immuno-responsive hydrogel-modulated RF resonant sensor for capturing and amplifying virus antigen recognition, the wireless immunoassay achieves simultaneous detection of three types of respiratory viruses. Direct detection of clinical samples shows high accuracy in diagnosing respiratory virus infections. This work provides a sensitive and accurate immunoassay technology for on-site virus detection and disease diagnosis, compatible with wearable integration. The study is well-designed and comprehensive. However, some questions need to be considered and clarified before acceptance.

Agreed.

We sincerely thank the reviewer for the patience and time on reviewing our manuscript and gave us the thoughtful and constructive suggestions to improve the clarity and scientific rigor of our manuscript. We have carefully addressed the questions below and revised our manuscript accordingly.

Line 58, please avoid using etc. in the manuscript. Same for line 110, 507, 514

Agreed.

Thank the reviewer for the careful check. As the reviewer suggested, we have revised the text on Line 58, "...respiration behaviors such as talking, coughing, and sneezing."

On Line 110, we have revised the text as "...originates from respiratory tract (saliva and aerosols)".

On Line 507, we have revised the text as "...and exosomes in blood plasma^{22,25}"

On Line 514, we have revised the text as "(for example, infectious diseases and cancers)"

Line 99, facilitate□facilitates

Agreed.

Thank the reviewer for the careful check. As the reviewer suggested, we have revised the text on Line 99, "a paralleled RF readout network that facilitates robust and multiplexed detection".

Line 102, do you mean infection discrimination among these listed types of viruses or

infection detection or both? Please clarify.

Agreed.

Thank the reviewer for the kind suggestion. As the reviewer pointed out, we have included three sensors for detection of SARS-CoV-2, H1N1, and RSV, respectively. These three viruses are selected as they are ones of the representative viruses account for respiratory infections. Such design can realize the infection discrimination among these three viruses, However, co-infection of multiple viruses has also been reported recently, so we believe with the multiple sensors integrated, detection of co-infection of these viruses is also possible.

As the reviewer suggested, on Page 4, Line 100, we have revised the text as “As a demonstration, the wireless immunoassay technology manifests sensitive and specific detection of SARS-CoV-2, influenza A H1N1, and RSV, for wearable and on-site monitoring of certain virus infection or co-infection of multiple viruses.”

Line 119, Will the breathing itself (absence of pristine antigen) also cause the swelling of the hydrogel to some extent? have you compared the swelling caused by the competitive binding of pristine antigen? How different are they?

Agreed.

Thank the reviewer for the insightful comment and kind suggestion. As the reviewer suggested, the swelling of the hydrogel can be divided into two parts: the non-specific swelling (inherent swelling in the absence of pristine antigen) and the specific swelling caused by competitive binding. Exactly like the reviewer assumed, aerosol itself from breathing can also cause the swelling of the hydrogel, as we have demonstrated in Fig. 3d-3f, Supplementary Fig. 23, and so on. In fact, evaluation of non-specific swelling is so important that we have added reference groups (aerosols generated from 1x PBS with no antigen) to all the experiments as comparison so we can tell whether the resonant frequency change is caused simply by moisture absorption or stimuli from pristine antigen. Please allow us to take the results from Supplementary Fig. 23 and Fig. 3e as examples to clarify this point, as shown below.

As shown in Supplementary Fig. 23, we have compared the sensor response in spiked aerosols and blank aerosols. All the aerosols were generated from 1x PBS as solution matrix and the only difference is the presence of pristine antigen. We used “NC” to term all the negative control groups in the absence of pristine antigen across the manuscript. As can be observed in Supplementary Fig. 23, the sensors showed much higher response in aerosols with very low concentration of pristine antigen than that in NC groups. For H1N1 sensor, the difference is quite obvious even within 1 min. On this basis, we are convinced that, although breathing itself can also cause the

swelling of hydrogel and sensor response to some extent, the competitive binding of pristine antigen can lead to more intensive swelling and obviously higher sensor response to be distinguished from normal breathing.

Supplementary Fig. 23 Rapid aerosol detection capability of the SARS-CoV-2 and H1N1 sensors. Response at 5 min is still significant enough to detect fg/L level virus antigen in aerosols, while it is more marginal for results at 1 min for low concentration virus detection.

Furthermore, considering the existence of the non-specific swelling, we determined that the sensor response in NC is the baseline of the sensor. We set the threshold as mean response in NC plus three folds of its deviation as the minimum detectable response caused by antigen competitive binding, as indicated by the dash lines in Fig. 3e. Signals above this threshold will be considered as antigen positive, and the limit of detections were calculated from the calibration plots. In this way, we make sure the detection of virus antigen is not misled by the non-specific swelling.

Fig. 3e Calibration plot of resonant response versus SARS-CoV-2 NP, H1N1 HA, and RSV FP concentration in spiked aerosol, respectively. The results were acquired at 10 min with 1× PBS as aerosol matrix.

We agree with the reviewer that it needs to be clarified, and made revisions on Page 13, Line 300. "The resonant signal can be divided into two parts: from non-specific swelling (inherent swelling in the absence of pristine antigen) and from specific swelling caused by competitive binding. To rule out the influence of non-specific swelling, negative control groups (aerosols generated from 1x PBS with no antigen,

termed as NC) has been added to all the experiments as comparison. Considering the existence of the non-specific swelling, the sensor response in NC is regarded as the baseline. The threshold of minimum detectable response is set to be mean response in NC plus three folds of its deviation, as indicated by the dash lines in Fig. 3e. Signals above this threshold are considered as virus positive, and the limit of detections were calculated from the calibration plots.”

Line 120, is that aggregation occurred inside the preformed hydrogel? And why the competitive antigen binding leads to severe AuNP aggregation? Is this triggered by the amount of the virus/spike proteins on the surface? Please clarify.

Agreed.

Thank the reviewer for the insightful comment and kind suggestion. As the reviewer assumed, the aggregation occurred inside the preformed hydrogel. Before the competitive binding, the AuNPs-Ab were immobilized within the hydrogel by immuno crosslinks. After the exposure to virus particles/fragments/proteins, these AuNPs-Ab tend to bind with the pristine proteins and form aggregates, as we have demonstrated in Supplementary Fig. 9c. For the second question, as the reviewer suggested, the aggregation is triggered by the virus particles that contain abundant binding sites to lead these AuNPs-Ab to bind together.

Supplementary Fig. 9c SEM images and element mapping of the aggregated AuNPs-Ab. Scar bar: 1 μ m. The focused distribution of Au indicates the existence of AuNPs, while the focused N distribution is attributed to the encapsulation of antigen/antibody proteins.

As the reviewer suggested, we have clarified the mechanism of this aggregation

on Page 5, Line 118-123 as “Thus, exposure of pristine antigen (virus stimuli) can lead to the competitive binding of AuNPs-Ab to these targets in the preformed hydrogels, breaking the immuno crosslinks and releasing the hydrogel networks. Meanwhile, the abundant binding sites on virus particles lead these AuNPs-Ab to bind together and form aggregates (Supplementary Fig. 3a-3b and Supplementary Fig. 8c). Due to the LSPR effect, the blue color of hydrogels gradually fades out (Supplementary Fig. 3c).”

Line 132, when you refer to antigen stimuli, is it carried out by immersing in a solution or through an actual breathing test? And the stimuli for how long? Minutes or hours? Please clarify.

Agreed.

Thank the reviewer for the insightful comment kind suggestion. On Line 132, the description is linked to the Fig. 1c and Supplementary Fig.5a-b, which were carried out by immersing in a solution. The stimuli lasted for 3 hours before the acquisition of the SEM images.

As the reviewer suggested, we have clarified the description on Line 132 as “The network of immuno-responsive hydrogel exhibited expansion after 3-hours-of antigen stimuli in 1 ng/mL spiked solution (Fig. 1c and Supplementary Fig.5a-b).”

On Line 148, the caption has been modified as “Scanning electron microscopy (SEM) images of hydrogels before (top) and after immersing in 1 ng/mL spiked solution for 3 hours.”

Line 148: Scar bar?? Do your mean scale bar? Same for 150 and other places in the manuscript

Agreed.

Thank the reviewer for the careful check. We are so sorry for the typos. As the reviewer suggested, we have revised them on Line 148, 150, and 192 of the manuscript, and on Line 102, 116, 118 of the Supplementary information.

Line 157: “Infection symptom monitoring” is quite broad. Do you mean monitor the body temperature? Need to be very specific about what you are trying to monitor.

Agreed.

Thank the reviewer for the comment and kind suggestion. We are so sorry for the ambiguity. As the reviewer pointed out, we included a temperature sensor to monitor the potential fever (breath temperature rise) symptom accompanied with the virus infection.

As the reviewer suggested, we have clarified it on Page 6, Line 157 “As a

demonstration, three ImmHR sensors targeted for contagious respiratory virus (SARS-CoV-2, influenza A H1N1, and RSV) and a temperature sensor for monitoring of breath temperature have been integrated, which aims to provide fast virus discrimination and monitoring of potential fever symptom”.

Line 182: “which also leads to AuNPs aggregation” is this aggregation caused by the competitive immunobinding? Or swelling? Or both? This question also reflects my previous comments on has the author carried out the control experiment of the swelling test of the hydrogel itself without the virus treatment?

Agreed.

Thank the reviewer for the insightful comment and kind suggestion. As far as we are concerned, the aggregation is mainly due to the competitive immunobinding. From the perspective of sensing mechanism, the AuNPs-Ab will detach from the polymer chains only when exposed to pristine antigen which has higher binding affinity, thus it is this competitive immunobinding that accounts for the AuNPs aggregation. As we illustrated above, the abundant binding sites on virus particles tend to bind these AuNPs together. Besides, the multiple protein-protein and protein-AuNPs interactions such as hydrogen bond, electrostatic interaction, and gold-sulfur interaction will also contribute to the assembly of AuNPs-Ab when they are detached from the polymer chains and form aggregates. The non-specific swelling in control experiments, however, should not lead to this chemical transition as it only absorbs more moisture to lead to minor volume change.

For the second question, as we answered above, yes we have carried out control experiment of the swelling without virus treatment, termed as NC across the manuscript. And it showed that these NC groups demonstrate much lower swelling and sensor response compared to that with pristine antigen treatment.

As the reviewer suggested, we have clarified the mechanism of the AuNPs aggregation on Page 7, Line 182 “Further competitive immuno binding between the virus antigen and AuNPs-Ab triggers intense hydrogel swelling owing to the breaking of immuno crosslinks. The multiple protein-protein and protein-AuNPs interactions such as hydrogen bond, electrostatic interaction, and gold-sulfur interaction will also contribute to the assembly of AuNPs-Ab when they are detached from the polymer chains and form aggregates.”

Line 183: “the formation of immuno responsive hydrogel needs precise control”. Can the author please clarify how could this be achieved? Through reaction time? Temperature? The author may have specified this in figure 2a but also need to be point

out in the main content.

Agreed.

Thank the reviewer for the comment and kind suggestion. As the reviewer pointed out, the precise control of the hydrogel formation is realized through reaction time. The temperature is also important yet we make sure it is performed under room temperature (~25 °C) so this parameter has been ruled out.

As the reviewer suggested, we have specified it in the main content on Page 7, Line 183 “Notably, the formation of immuno responsive hydrogel needs a precise two-step control: the first step is a quick copolymerization of AAm and vinyl-antigen with APS and TEMED within 60 s, while the second step is crosslinking of the polymerized antigens with MBAA and AuNPs-Ab within 5 min to realize solidified hydrogels. Notably, inadequate copolymerization would fail to immobilize AuNPs-Ab in the following crosslinking (it produces AuNPs-Ab-antigen precipitate before the hydrogel formation), while excessive copolymerization would lead to direct formation of hydrogels free of immuno crosslinks.”

On Page 23, Line 555, we have made the revision “The synthesis of AuNPs-Ab crosslinked hydrogels follows a two-step gelation strategy under room temperature (~25 °C).”

Line 216: “Although the red, green and blue parameters show similar trend in colorimetric assays” can you please define what are the red, green and blue parameters? If this is related to the aggregation state of the AuNPs then this needs to be clarified. I realise some of this information went into the method section from line 650, please consider to move some of those to the main text to eliminate confusion.

Agreed.

Thank the reviewer for the comment and kind suggestion. As the reviewer pointed out, the red, green and blue parameters refer to the hydrogel color that related to the aggregation state.

As the reviewer suggested, we have moved the information to main text on Page 9, Line 216 “The as-prepared hydrogels demonstrate blue color, which gradually fades out due to AuNPs-Ab aggregation in the presence of antigen or virus stimuli. Thus, the colorimetric assay has been used to evaluate the responsiveness of the hydrogels to optimize the composition. Images of hydrogels before/after reaction were obtained by a camera. Although the red, green, and blue parameters of the Image J processed hydrogel images showed similar trend after reaction, the change of red parameter (ΔR) presents a lower standard deviation (σ) and thus been selected for further study (Supplementary Fig. 9).”

Line 269: “ImmHR sensors with 700 μm thick hydrogel are determined for reliable269 immunoassays in following study” have you considered the drying effect to the immune response and the resonant response of a 700 μm thick hydrogel? Or the hydrogel must be maintained moisturised during the whole process? Please specify this.

Agreed.

Thank the reviewer for the insightful comment and kind suggestion. As the reviewer pointed out, we have considered the drying effect to the immune response and the resonant response of our hydrogel. As exhibited in Supplementary Fig. 25, we have assessed the influence of preservation humidity on hydrogel dehydration, and it showed that even under 98% RH, the hydrogel underwent slow drying as can be identified by the mass loss. Such results reminded us that the hydrogel sensor must be stored in highly humid environment to avoid the dramatic loss of sensing capability. And as the reviewed assumed, the hydrogels were stored in refrigerator (98% RH, 4 $^{\circ}\text{C}$) before use. However, considering that the sensor is aimed to detect in aerosols or solution, both of which provide highly moisturized environment, such decay of performance during the use is inhibited.

For further clarification, the stability issue is a general challenge to hydrogels.¹ Our hydrogel-based devices can attain 89.1% of the detection performance compared to the as-prepared ones (Supplementary Fig. 26). However, obvious decline of sensing performance has been observed after a week. We attribute it to the slow dehydration effect of the gel and the inactivation of antibodies. It might be improved by the chemical modification of polymers to achieve low volatility gels,² and careful tuning of the crosslink strength to sustain hydrogel’s sensing performances.¹

As the reviewer suggested, on Page 11, Line 268, we have specified the storage condition “However, thinner hydrogels are more susceptible to manufacturing fluctuations, possibly due to the easier dehydration effect during the process that undermines their sensing performances. Thus, ImmHR sensors with 700 μm thick hydrogel are determined for reliable immunoassays in following study, stored in moisture environment (98% RH in refrigerator) to avoid drying.”

On Page 16, Line 370, we have specified the stability issue “Nevertheless, similar to other hydrogel-based devices, the stability issue is a major challenge when it comes to scalable deployment of this wireless immunoassay device. It might be improved by the chemical modification of polymers to achieve low volatility gels,¹ and careful tuning of the crosslink strength to sustain hydrogel’s sensing performances.²”

Supplementary Fig. 25 Influence of preservation humidity on hydrogel dehydration. The RH is modulated by storing the hydrogels in the top air of saturated saline solution (Na₂HPO₄, RH =98%; KCl, RH=85%, NaCl, RH=76%).

Supplementary Fig. 26 Stability of the ImmHR sensors in two weeks. The responses were tested with H1N1 sensors in 71.4 pg/L H1N1 antigen aerosols. The hydrogel devices were stored in 98% RH and 4 °C.

Reference:

1. Zhang, K., Feng, Q., Fang, Z., Gu, L. & Bian, L. Structurally Dynamic Hydrogels for Biomedical Applications: Pursuing a Fine Balance between Macroscopic Stability and Microscopic Dynamics. *Chem Rev* 121, 11149-11193, (2021).
2. Wang, B. et al. Wearable bioelectronic masks for wireless detection of 783 respiratory infectious diseases by gaseous media. *Matter* 5, 4347-4362, (2022).

line 568: please change “frozen dried” to freeze-dried

Agreed.

Thank the reviewer for the careful check. We are so sorry for the typos. As the reviewer suggested, we have revised it on Line 568 “The hydrogels were freeze-dried and observed by scanning electron microscope (GEMINI 300, Zeiss, Germany).”

We thank the reviewer again for the careful check across the manuscript. We have checked through the manuscript and made revisions to the typos and spelling mistakes.

REVIEWERS' COMMENTS:

Reviewer #1 (Remarks to the Author):

Even though I still think the robustness of the sensor is not so good. Overall, I think this work is comprehensive and impressive. The authors have addressed most of my concerns, I suggest to accept this work.

Reviewer #2 (Remarks to the Author):

Thanks for your answers. Please, check again Supplementary Figure 17 since the x-axis seems to be counterintuitive and may be wrong.

Reviewer #3 (Remarks to the Author):

The authors have addressed all questions thoroughly, and I have no further questions. The paper should be accepted for publication at this stage.

Response to comments of Reviewers

Reviewer #1 (Remarks to the Author):

Even though I still think the robustness of the sensor is not so good. Overall, I think this work is comprehensive and impressive. The authors have addressed most of my concerns, I suggest to accept this work.

Agreed.

Thank the reviewer for your insightful comments to improve this manuscript and the positive conclusion about our work. We agree with the reviewer that the robustness of the sensor is a major challenge, as the alignment and wireless reading can be interfered by many operation errors. Nevertheless, as the reviewer indicated, it does not hamper the comprehensiveness and impressiveness of this work. We look forward to more endeavors possibly inspired by this work to solve the challenges about robustness in the future.

Reviewer #2 (Remarks to the Author):

Thanks for your answers. Please, check again Supplementary Figure 17 since the x-axis seems to be counterintuitive and may be wrong.

Agreed.

Thank the reviewer for your careful check and kind suggestion. We are so sorry about the error of x-axis in Supplementary Figure 17. As the reviewer suggested, the x-axis should not be continuous as the bending radii of flat device equals to infinite value. We have revised Supplementary Figure 17, as shown below.

Thank you again for your valuable time and efforts to help us improve this work.

Supplementary Fig. 17 Influence of bending radii on sensor readout.

Reviewer #3 (Remarks to the Author):

The authors have addressed all questions thoroughly, and I have no further questions. The paper should be accepted for publication at this stage.

Agreed.

We sincerely thank the reviewer for your valuable efforts to review this manuscript and the positive conclusion made.